Methods

# Cell-Int: a cell–cell interaction assay to identify native membrane protein interactions

Thibaud Aymoz-Bressot[1], Marie Canis[1,2], Florian Meurisse[3], Anne Wijkhuisen[4] (ORCID), Benoit Favier[3] (ORCID), Guillaume Mousseau[2], Anne Dupressoir[1], Thierry Heidmann[1,2], Agathe Bacquin[2] (ORCID)

**Intercellular protein–protein interactions (PPIs) have pivotal roles in biological functions and diseases. Membrane proteins are therefore a major class of drug targets. However, studying such intercellular PPIs is challenging because of the properties of membrane proteins. Current methods commonly use purified or modified proteins that are not physiologically relevant and hence might mischaracterize interactions occurring in vivo. Here, we describe Cell-Int: a cell interaction assay for studying plasma membrane PPIs. The interaction signal is measured through conjugate formation between two populations of cells each expressing either a ligand or a receptor. In these settings, membrane proteins are in their native environment thus being physiologically relevant. Cell-Int has been applied to the study of diverse protein partners, and enables to investigate the inhibitory potential of blocking antibodies, as well as the retargeting of fusion proteins for therapeutic development. The assay was also validated for screening applications and could serve as a platform for identifying new protein interactors.**

## Introduction

Intercellular protein–protein interactions (PPIs) are essential for cellular cohesion and communication in multicellular organisms. They are involved in a variety of biological processes including cell proliferation and differentiation, immune response, host–pathogen interaction, synaptic transmission, and fertilization.

During fertilization, the interaction between both gametes and the subsequent fusion step heavily rely on intercellular PPIs. In mammals, the interaction between the egg membrane protein JUNO and the sperm membrane protein IZUMO1 is essential for sperm–egg binding and hence for fertilization (Inoue et al, 2005; Bianchi et al, 2014). In the field of immunology, cellular and molecular studies have also shown over the years the importance and

complexity of the cell–cell contacts involved in the immune response, leading to the identification of a series of membrane proteins shaping immune cell signaling, activation, and function. Among these proteins, immune checkpoint proteins, notably PD-1 and CTLA-4, and their ligands have been extensively studied for their contribution to the fine-tuning of immune responses, as well as their therapeutic interest in immuno-oncology (Gaikwad et al, 2022). Regulation of immune cell activation also involves a family of paired receptors sharing similar extracellular ligand-binding regions but having either inhibitory or activating signaling functions (Kuroki et al, 2012). Inhibitory receptors (e.g., LILRB1-5, KIR2DL1-4, PILRα, CD200R1) display one or several immunoreceptor tyrosine-based inhibitory motifs in their cytoplasmic region, whereas activating receptors usually associate with an adaptor protein bearing an immunoreceptor tyrosine-based activation motif (e.g., LILRA1-2, KIR2DS1-5, PILRβ). These receptors often interact with one or several self-ligands such as MHC class I molecules. One such ligand is the non-classical MHC-I molecule HLA-G, which plays a major role in maternal–fetal tolerance enabling the embryo and placenta, considered as semi-allogeneic, to be tolerated by the maternal immune system throughout pregnancy (Ferreira et al, 2017). HLA-G has at least seven distinct isoforms, named HLA-G1–7, of which four are membrane-bound (HLA-G1–4) and three are secreted in a soluble form (HLA-G5–7) (Carosella et al, 2015). Among these isoforms, only HLA-G1 and its soluble counterpart HLA-G5 can associate with β2-microglobulin (β2m), such as classical MHC-I molecules (Morales et al, 2003). HLA-G exerts its immunosuppressive activity by binding to receptors expressed on the surface of different immune subpopulations, such as T and B lymphocytes, NK cells, neutrophils, and dendritic cells. Five receptors have already been described in the literature, with, in particular, the immunoreceptor tyrosine-based inhibitory motif–containing LILRB1, LILRB2, and KIR2DL4 receptors (Carosella et al, 2015). HLA-G expression is typically limited to the placenta and other immune-privileged sites in a physiological context, but can become dysregulated in certain pathological conditions, including various

---

[1]CNRS UMR9196, Laboratory of Molecular Physiology and Pathology of Endogenous and Infectious Retroviruses, Gustave Roussy, Université Paris-Saclay, Villejuif, France [2]VIROXIS, Gustave Roussy, Villejuif, France [3]Université Paris-Saclay, Inserm, CEA, Center for Immunology of Viral, Auto-immune, Hematological and Bacterial Diseases (IMVA-HB/IDMIT), Paris, France [4]Université Paris-Saclay, CEA, INRAE, Médicaments et Technologies pour la Santé (MTS), Gif-sur-Yvette, France

Correspondence: agathe.bacquin@gustaveroussy.fr
Marie Canis's present address is Laboratory of Retrovirology, The Rockefeller University, New York, NY, USA

types of tumors (Amiot et al, 2011). The immunosuppressive properties of HLA-G are believed to facilitate tumor cell evasion of the immune system, making it a negative prognostic factor. As a result, HLA-G has been suggested as a new immune checkpoint molecule (Carosella et al, 2015; Krijgsman et al, 2020).

In virology also, PPIs between glycoproteins of enveloped viruses and receptors expressed at the membrane of target cells are critical for viral infection. Several viral glycoprotein/host receptor pairs have been identified during the past three decades, some viruses, such as measles virus (MeV), having acquired several cell surface receptors during evolution (Laksono et al, 2016). Endogenous viral glycoproteins such as Syncytins, which are essential to mammalian placentation via the formation of syncytial cell layers at the fetal–maternal interface (Dupressoir et al, 2012), also interact with one or several membrane receptors expressed on specific cell subsets. The identification of virus–receptor interactions and their characterization are a key to understanding viral tropism and pathogenesis, as well as developing novel antiviral therapeutics targeting viral entry.

Overall, because of their accessibility and functional importance, plasma membrane proteins are excellent therapeutic targets, accounting for more than 60% of drug targets (Overington et al, 2006), while constituting about 23% of the human proteome (Uhlén et al, 2015). Among the drugs developed against these proteins, small molecules are widely used in therapeutics. Antibodies and antibody-like molecules are also an important class of newly developed drugs targeting membrane proteins. In oncology, immunotherapy is now being proposed as a first-line treatment for some malignancies (Ooki et al, 2021; Reck et al, 2022) and mainly relies on the inhibition of membrane PPIs involved in immune down-modulation. The well-known PD-1, PD-L1, or CTLA-4 immune checkpoints are among the most targeted. Blocking these inhibitory receptors or ligands restores immune cell function and therefore antitumor immunity. Antibodies also target other membrane proteins in order to inhibit the activating signals they deliver to tumor cells. The well-characterized cetuximab, approved for the treatment of colon and head and neck cancers, prevents EGFR signaling by blocking the interaction with its activating ligands (Ciardiello & Tortora, 2008).

Thus, methods to investigate membrane PPIs and how drugs inhibit them are essential. However, intercellular PPIs between membrane proteins are challenging to study: because of the amphipathic nature of membrane proteins, classical approaches to study PPIs are more difficult to apply compared with soluble proteins. Detergent solubilization of these membrane-embedded proteins can disrupt their native conformation. In addition, extracellular PPIs involved in signaling networks are often weak and transient (Bagheri et al, 2020) and may require adequate post-translational modifications, such as glycosylation and disulfide bonds, to be detected. Current experimental methods to study extracellular PPIs often use purified and/or modified proteins (e.g., ectodomains of membrane proteins, tagged proteins) and characterize the interaction under non-physiological conditions.

Here, we describe Cell-Int: a cell–cell interaction assay allowing the study of intercellular interaction between two membrane proteins under quasi-physiological conditions. This technique was first validated using different known protein partners. Then, we

show the usefulness of Cell-Int as a tool to screen for membrane partners among a library of ORFs. With the example of HLA-G, the technique was used to investigate the interaction of all membrane-bound HLA-G isoforms with receptors, thereby providing further information on the interaction network of these proteins. Finally, using competitive experiments, we show that Cell-Int is suitable to study the inhibitory potential of blocking antibodies, as assessed by $IC_{50}$ values, which are potent indicators of the therapeutic efficiency of developed drugs.

# Results

### Development of a cell interaction assay (Cell-Int) for intercellular PPI identification and characterization

In order to develop a simple versatile technique to study intercellular membrane protein interactions, we chose to express unmodified membrane proteins of interest in a suspension cell subtype, which can be efficiently transfected to produce high levels of proteins: the FreeStyle 293-F cell line (293-F). Two different batches of cells are transfected with the two membrane proteins to be tested, along with a different fluorescent protein (GFP or RFP), to visualize each batch of cells by flow cytometry and indirectly measure the transfection efficiency (Fig 1A). We made the assumption that the overexpression of two interacting partners at the cell surface will induce the formation of cell aggregates upon co-incubation. These aggregates can be visualized as double-positive events by flow cytometry. As a proof of concept, we started with two membrane interacting partners playing a key role in egg–sperm binding: Juno and Izumo1 (Inoue et al, 2005; Bianchi et al, 2014) (Fig 1B–E).

To determine the optimal temperature for cell interaction, we tested three usual temperatures in protein binding studies: 4°C, 25°C, and 37°C (Fig 1B and C). At 4°C and 25°C, although the background of double-positive events among transfected cells was very low among the three control conditions transfected with an empty vector (e.v.) (Fig 1B, lower left dot plot, and Fig 1C), only the condition where Juno and Izumo1 were both expressed showed an increase in the percentage of double-positive events (Fig 1B, lower right dot plot, and Figs 1C and S1A). However, the increase was only statistically significant at 25°C. Importantly, the interaction signal does not depend on the nature of the co-expressed fluorescent protein, as the Juno/Izumo1 interaction signal was similar when Juno was co-expressed with GFP and Izumo1 with RFP, compared with the opposite combination (Fig S1B). At 37°C, the percentage of double-positive events was also significantly increased when Juno and Izumo1 were both overexpressed, although the background signal was high in the control conditions. Experiments using other protein partners yielded comparable results (data not shown), leading us to conserve a temperature of 25°C during cell co-incubation for further testing. To determine the kinetics of cell interaction in this assay, we co-incubated both batches of cells expressing Juno or Izumo1 together for up to 3 h at 25°C and analyzed the percentage of double-positive events at several time points (Fig 1D). As early as 10 min post–co-incubation of the two cell

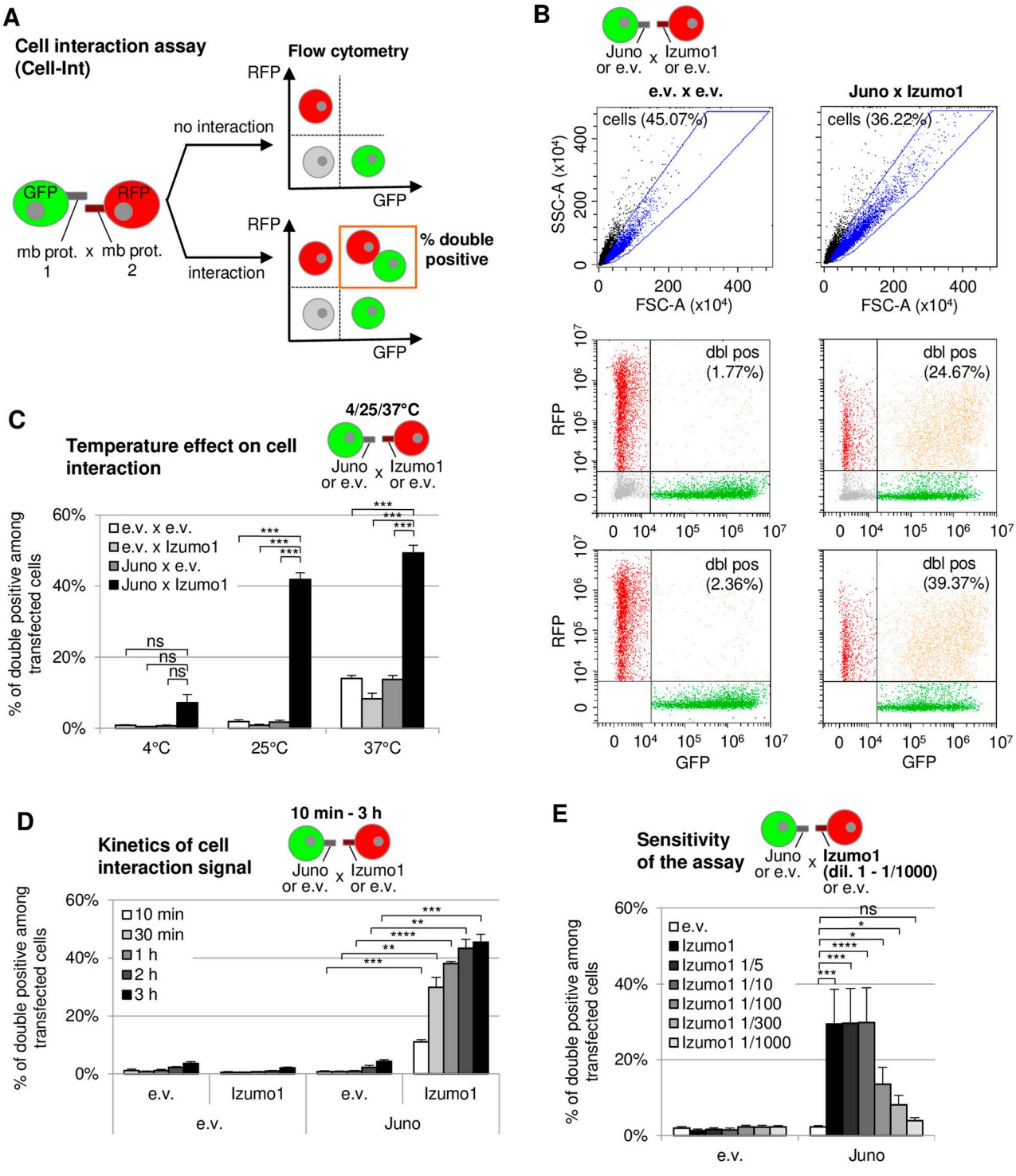

**Figure 1. Cell interaction assay (Cell-Int) to detect extracellular interactions between native membrane proteins.**
**(A)** Schematic representation of Cell-Int. Two different batches of 293-F cells are co-transfected with GFP or RFP, as well as one of the two membrane protein (mb prot.) expression vectors. An increase in the percentage of GFP and RFP double-positive events measured by flow cytometry after co-incubation of the two cell batches indicates cell–cell interactions induced by membrane protein interactions. **(B, C, D, E)** Juno/Izumo1 interaction was assessed using Cell-Int by transfecting one batch of 293-F cells (in green) with GFP and Juno expression vectors or an empty vector control (e.v.), and a second batch (in red) with RFP and Izumo1 expression vectors or an e.v. control. **(B)** Cells (blue dots) are gated on SSC-A = f(FSC-A) dot plot ((B), top panel); then, the percentage of double-positive events (dbl pos; orange dots) is quantified ((B), bottom panel) after exclusion of non-transfected double-negative cells (gray dots) on the RFP = f(GFP) dot plot ((B), middle panel). **(C, D, E)** Quantification of the cell–cell interaction is represented on bar graphs displaying the percentage of double-positive events among transfected cells (means ± SEM). Statistics were performed using a paired $t$ test (four independent experiments; ns, $P > 0.05$; *$P \leq 0.05$; **$P \leq 0.01$; ***$P \leq 0.001$; ****$P \leq 0.0001$). **(C)** Three different temperatures were tested for the co-incubation of the two cell batches for 1 h: 4°C, 25°C, and 37°C; interaction signals were measured between GFP and RFP cells both transfected with an e.v. (white

batches was the percentage of double-positive events significantly increased compared with the control condition. The signal further increased over time, going from 11% of double-positive events at 10 min to 45% at 3 h. However, if the background noise of the control condition is subtracted, the interaction signal reaches a plateau between time points 1 and 2 h and remains relatively stable until at least 3 h of incubation time (Fig S1C). These findings were corroborated when other known protein partners were tested (data not shown). In view of the kinetics, we chose to analyze cell interaction signals after 1–2 h of incubation time in subsequent experiments.

To estimate the number of cells composing the aggregates, two complementary strategies were employed: flow cytometry analysis of double-positive events after Hoechst staining and cell sorting of double-positive events, followed by fluorescence microscopic analysis. The results of the Hoechst staining demonstrated that most of the aggregates were doublets, comprising over 40% of the total, whereas triplets and quadruplets represented 12–19% of the aggregates. The remaining portion of the aggregates consisted of those with five or more cells (Fig S1D and E). The microscopic analysis yielded results that were consistent with those of the Hoechst staining analysis. Specifically, 50% of the aggregates were identified as doublets, 17% as triplets, 14% as quadruplets, and the remainder as aggregates comprising five or more cells (Fig S1E and F). It is worth noting that approximately one-third of the aggregates also contained one or more non-fluorescent cells (data not shown). This suggests that they are either not transfected or express only the membrane protein. As a potential avenue for further optimization, the membrane proteins of interest and the fluorescent markers could be expressed on a single expression vector. However, the use of a vector expressing *Izumo1* and *RFP* under the same promoter with an IRES did not result in enhanced interaction signals (data not shown). Furthermore, this approach would reduce modularity as it would require additional molecular cloning.

The prospect of using this assay for screening purposes prompted us to check the sensitivity of the assay when one of the membrane proteins is expressed at low levels. To test it, Izumo1 expression vector was transfected at various dilutions ranging from 1 to 1/1,000th (Fig 1E) and the level of the expression of Juno and Izumo1 was examined for each condition (Fig S2A and B). The interaction signal indicated by the double-positive events was significant at dilutions 1 to 1/300th.

Altogether, our results suggest that the cell interaction assay, which we have named Cell-Int, is able to specifically detect cell-to-cell interaction induced by extracellular PPIs.

### Cell-Int is able to detect intercellular PPIs involved in various biological processes

As Juno and Izumo1 interaction is critical for mammalian egg–sperm binding, we wondered whether Cell-Int is also able to detect receptor–ligand interactions involved in immunoregulatory functions. To do so, we studied a series of receptor–ligand pairs (Figs 2A and S3A–I). In particular, we tested the checkpoint protein PD-1 and its binding partners PD-L1 and PD-L2, the other checkpoint protein CTLA-4 with CD80, and several receptors belonging to the family of paired receptors, also involved in immunoregulation: CD200R1 and the related protein CD200R1L, with CD200; and the inhibitory receptor PILRα with two of its binding partners NPDC-1 and COLEC12, and the corresponding activating receptor PILRβ with the same two ligands. Cell-Int results were in line with previous studies, giving significant interacting signals compared with control conditions for expected binding pairs, that is, PD-1/PD-L1, PD-1/PD-L2, CTLA-4/CD80, CD200/CD200R1, PILRα/NPDC-1, and PILRα/COLEC12 (Linsley et al, 1991; van der Merwe et al, 1997; Freeman et al, 2000; Latchman et al, 2001; Tseng et al, 2001; Wright et al, 2003; Sun et al, 2012; Cheng et al, 2013), whereas other related negative control pairs did not (CD200/CD200R1L, PILRβ/NPDC-1, and PILRβ/COLEC12), although CD200R1L and PILRβ were well expressed at the cell surface (Fig S3G and I). We noted that the PILRβ/NPDC-1 pair gave a significant increase compared to the control condition with PILRβ expressed only, but the signal was similar to the control with only NPDC-1 expressed. This signal could be due to the interaction of NPDC-1 with an endogenously expressed ligand.

To check whether Cell-Int is also relevant for the characterization of virus–receptor interaction, we tested a number of viral glycoprotein/receptor pairs identified in previous studies from our laboratory and others (Fig 2B): SARS-CoV-2 Spike (CoV-2 Spike) with its receptor ACE2 (Zhou et al, 2020), measles virus (MeV) hemagglutinin glycoprotein (H; Edmonston strain) with its three receptors Nectin-4, CD46, and SLAM (Dörig et al, 1993; Naniche et al, 1993; Tatsuo et al, 2000; Navaratnarajah et al, 2008; Mühlebach et al, 2011; Noyce et al, 2011), and three endogenous retroviral proteins Syncytin-2, Syncytin-A, and GLN Env, and their respective receptors MFSD2, Ly-6E, and Slc19A1 (Esnault et al, 2008; Bacquin et al, 2017; Tsang et al, 2019). The viral proteins mentioned above have been described as having multimeric structures, with MeV H being tetrameric, whereas SARS-CoV-2 Spike and Syncytins form trimers of dimers (Renard et al, 2005; Brindley & Plemper, 2010; Ruigrok et al, 2019; Huang et al, 2020). Using our technique, we observed that the interaction signal of each viral protein/receptor pair was significantly increased compared with control conditions. Importantly, this demonstrates the assay's ability to measure interaction with multimeric proteins. Concerning Syncytin-A, it was interesting to note that the background signal in the control condition with the viral glycoprotein only was high. This is consistent with the endogenous expression of Ly-6E at high levels in 293T cells (Bacquin et al, 2017), from which the 293-F cells are derived. Furthermore, this background interaction is eliminated when endogenous Ly-6E is knocked out using the CRISPR/Cas9 technology (Fig S3J).

Retargeting of viruses for gene delivery or oncolytic virotherapy is an emerging strategy to achieve selective target cell infection. It can be done by modifying an existing viral glycoprotein, such as MeV H, to incorporate a ligand onto a membrane protein of interest

bar), transfected with an e.v. and Izumo1 expression vector, respectively (light gray bar), transfected with a Juno expression vector and an e.v. (dark gray bar), or transfected with Juno and Izumo1 expression vectors (black bar). **(D)** Cell interaction signal kinetics were followed by co-incubation of the two cell batches for up to 3 h at 25°C, with incubation times represented as shades of gray, from 10 min (white) to 3 h (black). **(E)** Assay sensitivity to dilution of one of the membrane protein expression vectors was tested by diluting the Izumo1 expression vector down to 1/1,000 (E) (shades of gray, with increasing dilutions from black to light gray); the interaction signal with the e.v. control–transfected RFP cells is in white.

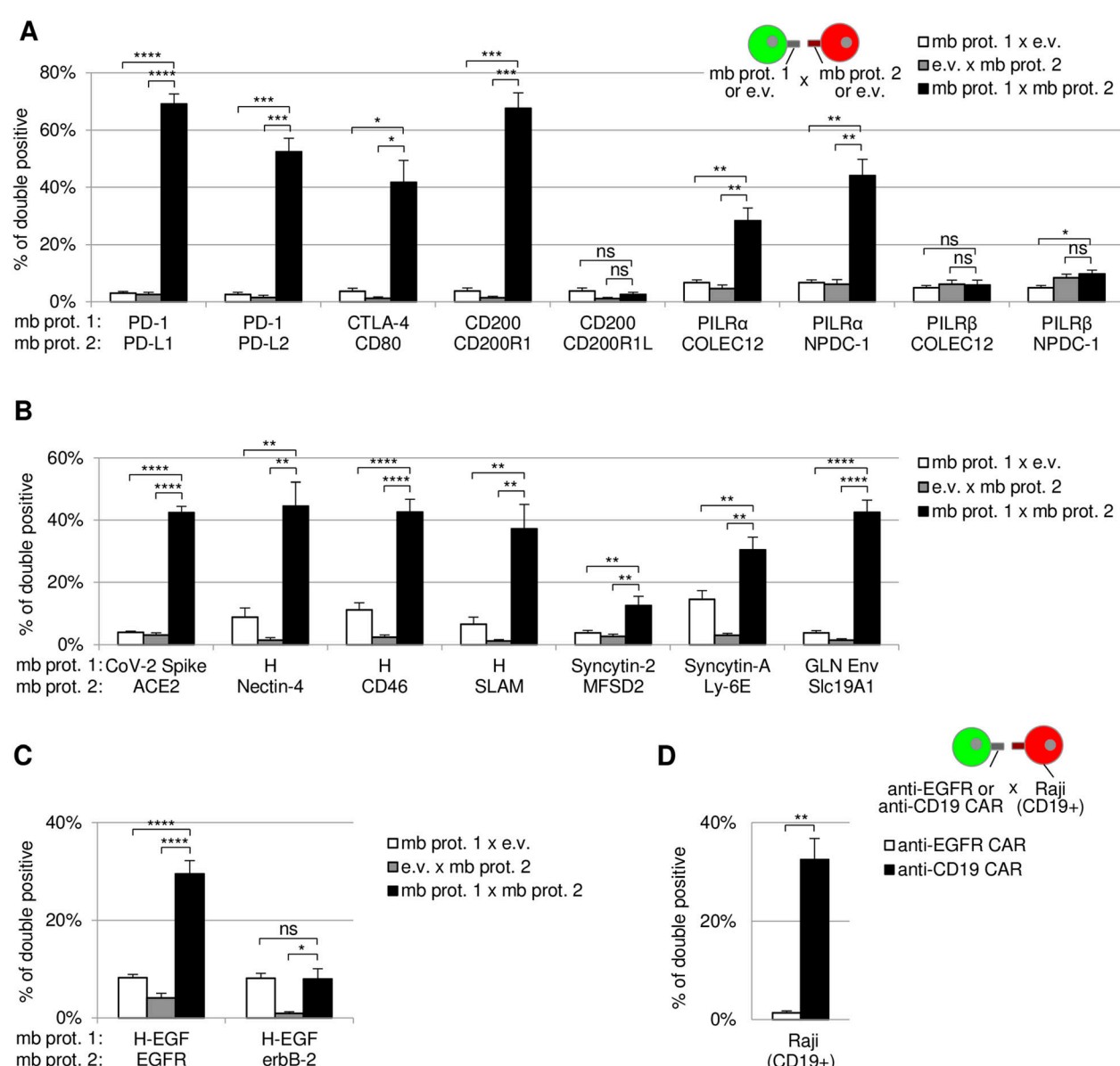

**Figure 2. Cell-Int specifically detects extracellular protein–protein interactions involved in various biological processes.**
**(A, B, C, D)** Cell interaction signals induced by membrane protein partners involved in (A) immunological pathways, (B) virus–receptor interactions, (C) virus retargeting strategies, or (D) chimeric antigen receptor–directed interaction are represented as bar graphs displaying the percentage of double-positive events among RFP-positive (A, B, C) or PE-positive (D) cells assessed by Cell-Int (means ± SEM). **(A, B, C)** Interaction signals were measured between GFP and RFP 293-F cells transfected with one of the membrane proteins (mb prot. 1) expression vector and an e.v., respectively (white bar), transfected with an e.v. and the second membrane protein (mb prot. 2) expression vector (gray bar), or transfected with each of the membrane protein expression vectors (black bar). **(D)** Interaction signals were measured between 293-F cells transduced with an anti-EGFR (white bar) or anti-CD19 (black bar) chimeric antigen receptor-IRES-GFP lentiviral vector, and Raji cells (CD19+) stained with a PE-coupled anti-CD20 antibody. **(A, B, C, D)** Statistics were performed using a paired $t$ test (3–10 independent experiments; ns, $P > 0.05$; *$P \leq 0.05$; **$P \leq 0.01$; ***$P \leq 0.001$; ****$P \leq 0.0001$).

(Navaratnarajah et al, 2012). In Fig 2C, we investigated whether Cell-Int could be used to test such retargeting strategies by examining the interaction between the fusion protein H-EGF (Funke et al, 2008) (Fig S3K) and its target EGFR (also known as erbB-1), or the related receptor erbB-2 as a negative control. The results showed a specific and significant interaction signal between the cells expressing H-EGF and those expressing EGFR and not with the related receptor erbB-2, although all proteins are well expressed at the cell surface (Fig S3L–N). This interaction between H-EGF and EGFR was due to EGF as it was not observed with H protein (Fig S3O). Similar to Syncytin-A, we observed that the background signal in the control condition with H-EGF only was also found to be elevated (Fig 2C). This background interaction signal was effectively eliminated when the interaction with endogenous EGFR was inhibited by the use of the anti-EGFR inhibitory antibody cetuximab (Fig S3P).

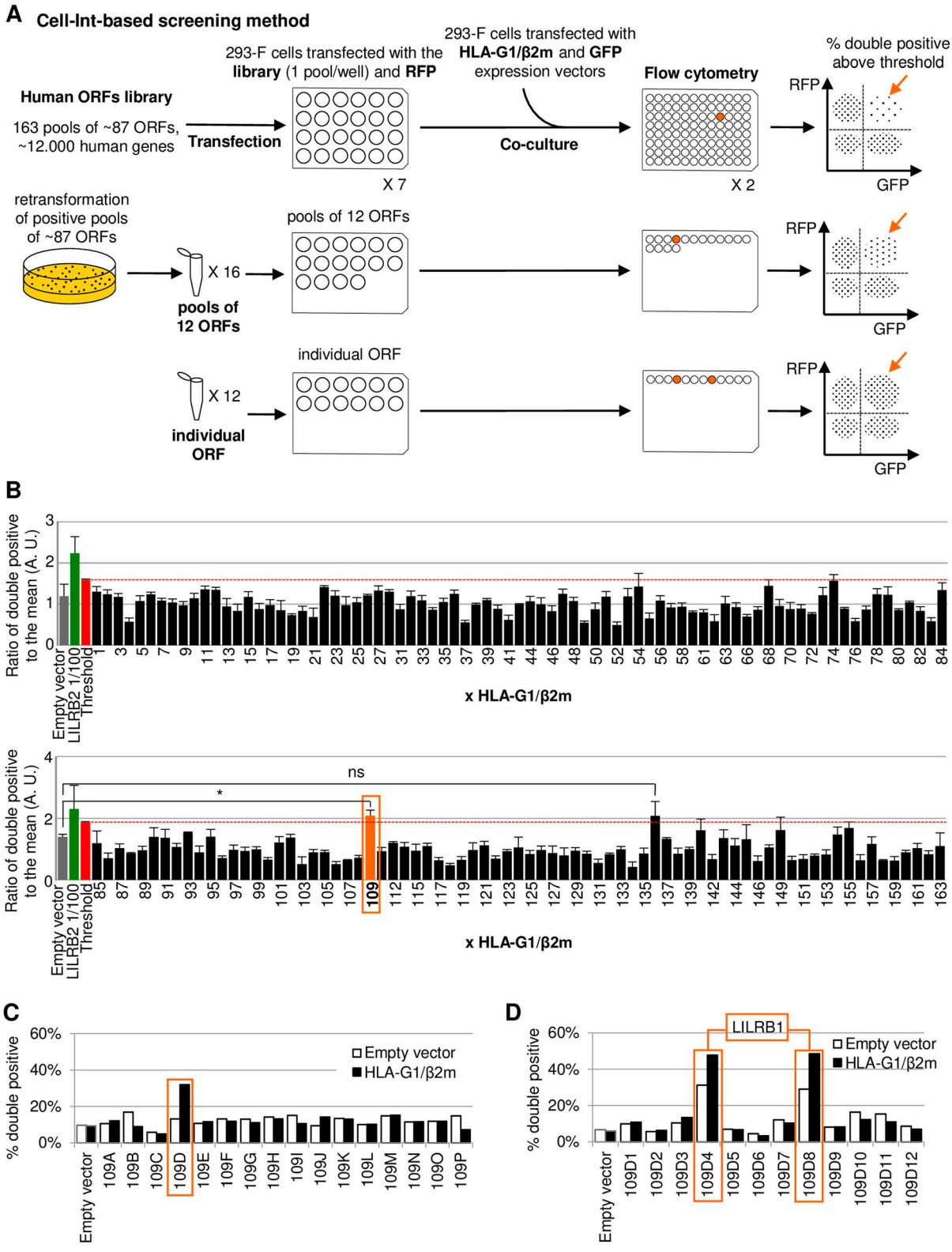

**Figure 3. Library screening using Cell-Int identifies LILRB1 as a ligand of HLA-G1.**
**(A)** Schematic of the screening procedure using an iterative approach to identify ligands inducing cell-to-cell interaction with HLA-G1/$\beta$2m- and GFP-co-transfected 293-F cells. **(B)** Bar graph showing the mean fold change (±SEM) in double-positive events induced by each cDNA pool–expressing RFP cells with HLA-G1/$\beta$2m- and GFP-expressing cells, normalized by the mean signal of the screened pools. The background signal (gray bar), a positive control (1/100[th] diluted LILRB2 expression vector; green bar), and the threshold value consisting of the mean signal plus two standard deviations (red bar, dashed red line) are indicated; negative cDNA pools are represented

T and NK cells engineered with chimeric antigen receptors (CARs) combining cell activation and antigen-binding domains are promising and expanding anticancer therapeutics, already approved for the treatment of hematological cancers (Sterner & Sterner, 2021), whereas applications on solid tumors need further improvement. Here, we tested whether our assay could be used to evaluate CAR-mediated cell targeting (Fig 2D). We showed that 293-F cells expressing anti-CD19 CAR were able to significantly bind the CD19[+] Raji cell line, whereas no interaction signal was observed when a control anti-EGFR CAR was expressed. Altogether, the technique has demonstrated its usefulness and sensitivity in the study of a wide range of interactions among membrane proteins with varying native structures and physiological functions.

### Cell-Int can be used for the screening of membrane partners

Identifying molecular interactors of a protein is a key to understanding its physiological importance, and screening of libraries is an efficient way to characterize such partners. We therefore set out to prove that Cell-Int could be useful for identifying membrane protein interactors in quasi-physiological conditions. To do so, we used the human ORF library ORFeome, covering about 12,000 genes of the human genome, to identify partners of the human protein HLA-G1. Given the time- and cost-effectiveness of this strategy, an iterative approach was used to screen 163 pools of 87 cDNAs on average (Fig 3A). Indeed, our dilution testing demonstrated that the interaction assay is sensitive enough to detect the interaction of a protein expressed from a cDNA diluted at 1:300 (Fig 1E). Each cDNA pool was co-transfected with an RFP expression vector, and their interaction with cells co-expressing HLA-G1, $\beta$2m, and GFP was measured by flow cytometry. Fig 3B shows the average normalized signals obtained from three independent screenings (non-normalized signals in Fig S4). The gray bar represents the background interaction (empty vector), and the green bar depicts the interaction between HLA-G1 and LILRB2 at a dilution of 1:100, which served as a positive control given that this receptor is not present in the ORFeome library. After identification and elimination of nine pools expressing proteins involved in cell–cell interaction and giving high interaction signals independent of HLA-G1 expression (Fig S4A), few cDNA pools led to an HLA-G1-specific interaction signal above the threshold value of mean plus two standard deviations (red bar, dashed red line), but only one, namely, pool 109, led to a significant interaction with HLA-G1 compared with the background interaction (Figs 3B and S4B). After two rounds of subcloning during which each pool giving a positive interaction signal was selected and subpooled (Fig 3A, C, and D), the individual ORF responsible for the interaction with HLA-G1 was sequenced, leading to the identification of LILRB1. This protein is a broad-range MHC class I receptor, and this may explain the increase in background interaction of the positive clones 109D4 and 109D8 with empty vector–transfected cells (white bars) because 293-F cells endogenously express classical MHC-I proteins. Overall, the

identification of a known partner of HLA-G1 by an unbiased screening of a cDNA library validates the usefulness of Cell-Int for protein–ligand screening.

### Application of Cell-Int to specify the interaction network of HLA-G1 with its described receptors

It is known that the suppressive activity of HLA-G1 on immune cells involves its interaction with membrane receptors, five of which have been described (Fig 4A, upper panel), with LILRB1 and LILRB2 having been more extensively studied (Carosella et al, 2015). Hence, we first analyzed whether these interactions could be observed using the cell interaction assay. As shown in Fig 4B, 293-F cells expressing RFP and LILRB1 or LILRB2 significantly interacted with GFP cells expressing HLA-G1/$\beta$2m (black bars) compared with an empty vector control (white bars). None of the other receptors, namely, KIR2DL4, CD160, or CD8, led to an interaction with HLA-G1/$\beta$2m despite being addressed to the plasma membrane (Fig S5A and B). Then, we investigated whether the presence of $\beta$2m was required for the interaction to establish. Results presented in Fig 4C indicate that HLA-G1–expressing 293-F cells (hatched bars) interacted to the same extent with LILRB1 (17.41%) and LILRB2 (8.56%) as do e.v.-expressing 293-F cells (19.93% and 9.48%, respectively, white bars). When $\beta$2m was co-transfected with HLA-G1 (Fig 4C, black bars), the interaction could occur between these cells and the ones expressing LILRB1 or LILRB2 (33.75% and 24.46%, respectively, $P <$ 0.01). Hence, our results show that $\beta$2m is necessary for HLA-G1 to interact with the two LILRBs.

To confirm our observations using a second, more conventional approach, we used a binding assay where 293-F–transfected cells were incubated with soluble Fc-tagged receptor ectodomains (Ed). As shown in Fig 4D and in line with the results from the cell interaction technique, a significant binding of LILRB1-Fc to HLA-G1/$\beta$2m-expressing 293-F cells (black bars) was detected, whereas there was no binding in the absence of $\beta$2m (hatched bars). Regarding LILRB2-Fc, no significant binding difference was observed between 293-F cells expressing either e.v, HLA-G1, or HLA-G1/$\beta$2m. We noted a high background interaction of this soluble receptor with 293-F cells, supposedly because of binding to other endogenously expressed HLA class I molecules. It is conceivable that the conformation of soluble LILRB2-Fc differs from that of the membrane-bound receptor, which could potentially result in increased binding to endogenously expressed molecules. This might explain the absence of differential binding of LILRB2-Fc between e.v- and HLA-G1/$\beta$2m-transfected 293-F cells. The same hypothesis may also be applicable to the case of LILRB1, which could explain the discrepancy in background binding of LILRB1 to e.v.-transfected cells between the cell-based assay and the binding assay. Among the other soluble receptors that we were able to produce, neither KIR2DL4-Fc nor CD160-Fc interacted with HLA-G1+/−$\beta$2m, thus confirming our observations with the cell interaction assay.

---

as black bars and the positive cDNA pool as an orange bar (three independent experiments; ns, $P > 0.05$; *$P \leq 0.05$, paired $t$ test). **(C, D)** Bar graphs showing the percentage of double-positive events formed between (C) subpooled candidates (109A-109P) or (D) individual clones (109D1-109D12) and e.v. (white bars)- or HLA-G1/$\beta$2m (black bars)-transfected cells.

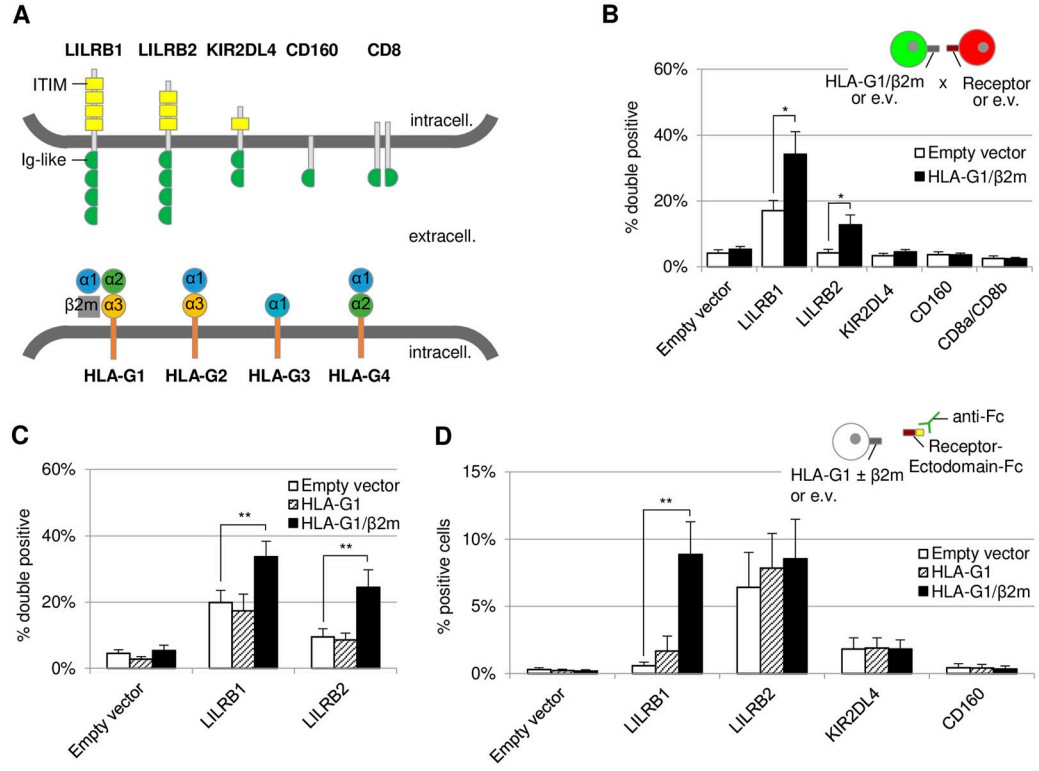

**Figure 4. Studying the HLA-G1 interaction network using Cell-Int.**
**(A)** Scheme of the five described HLA-G receptors (top) and the four membrane-bound HLA-G isoforms (bottom). Intracell., intracellular; extracell., extracellular. **(B)** Bar graph showing the percentage of double-positive events among RFP-positive cells, indicating cell interaction between receptor- and RFP-expressing 293-F cells and e.v. (white bars)- or HLA-G1/$\beta$2m (black bars)- and GFP-expressing 293-F cells, studied by Cell-Int. **(C)** Bar graph showing the percentage of double-positive events formed between RFP- and e.v.-, or LILRB1-, or LILRB2-expressing 293-F cells, and e.v. (white bars)- or HLA-G1 (hatched bars)- or HLA-G1/$\beta$2m (black bars)- and GFP-expressing 293-F cells, studied by the cell interaction assay. **(B, C)** Data are means ± SEM of eight to nine independent experiments. **(D)** Bar graph showing the binding of soluble Fc-tagged receptor ectodomains to e.v.-transfected (white bars) or HLA-G1–transfected (hatched bars) or HLA-G1/$\beta$2m-transfected (black bars) 293-F cells, as assessed by the percentage of anti-Fc–positive cells. Data are means ± SEM of five independent experiments. **(B, C, D)** Statistics were performed using a paired $t$ test (*$P \leq 0.05$; **$P \leq 0.01$).

## Cell-Int allows the study of HLA-G isoform interaction network

Besides HLA-G1, studies have shown that other HLA-G isoforms retain immunosuppressive activity, suggesting that their interaction with inhibitory receptors might be preserved (LeMaoult et al, 2013; Takahashi et al, 2016). Hence, we investigated, as shown in Fig 5, the interaction of the four described HLA-G membrane isoforms (HLA-G1–4, Fig 4A, bottom panel) with the five described receptors using Cell-Int. To avoid being hampered by the endogenous expression of human MHC-I proteins, we used two different cell lines: K562, which are human erythroleukemia cells that do not express MHC-I on their surface, and WOP cells, which are adherent murine fibroblast cells. We generated K562 and WOP cell populations stably expressing each of the four HLA-G isoforms and verified their expression by flow cytometry (Fig S5B and C). HLA-G1 was either expressed alone or co-transduced with $\beta$2m. As murine WOP cells do not express human $\beta$2m, we used $\beta$2m-transduced WOP cells as a negative control to account for any potential background interactions induced by $\beta$2m alone. The cell–cell interaction assay was hence performed between K562 or WOP cells expressing an HLA-G isoform and 293-F cells over-expressing one of the five receptors (Fig 5A and C). Our results in both cell lines show a significant interaction between cells

expressing LILRB1 or LILRB2 and cells expressing HLA-G1/$\beta$2m (black bars). None of the other receptors interacted with any of the HLA-G isoforms, and the LILRBs did not interact with HLA-G2, HLA-G3, or HLA-G4.

Interestingly, in contrast to what we observed in 293-F (Fig 4B) and WOP cells (Fig 5C), K562 cells overexpressing only HLA-G1 significantly interacted with 293-F cells expressing LILRB1 or LILRB2 (Fig 5A, hatched bars, $P < 0.01$), although to a lesser extent than HLA-G1/$\beta$2m-expressing K562 cells. This is probably due to endogenous HLA-G1–induced $\beta$2m up-regulation in this cell line, as shown by $\beta$2m staining (Fig S5B). Overall, these results are in agreement with our previous observations, showing the importance of $\beta$2m for the interaction of HLA-G1 with the LILRBs. Importantly, our experiments with the adherent fibroblastic WOP cell line show that Cell-Int can be adapted to adherent cells that once in suspension can establish interaction with other cells expressing their target ligand.

To confirm our results, we performed a binding assay where K562 or WOP cells were incubated with soluble His- or Fc-tagged receptor ectodomains, respectively (Fig 5B and D). Strong binding of LILRB1 and LILRB2 ectodomains to cells expressing HLA-G1/$\beta$2m was observed, whereas none of the other isoforms bound these receptors. These results are consistent with the cell–cell interaction

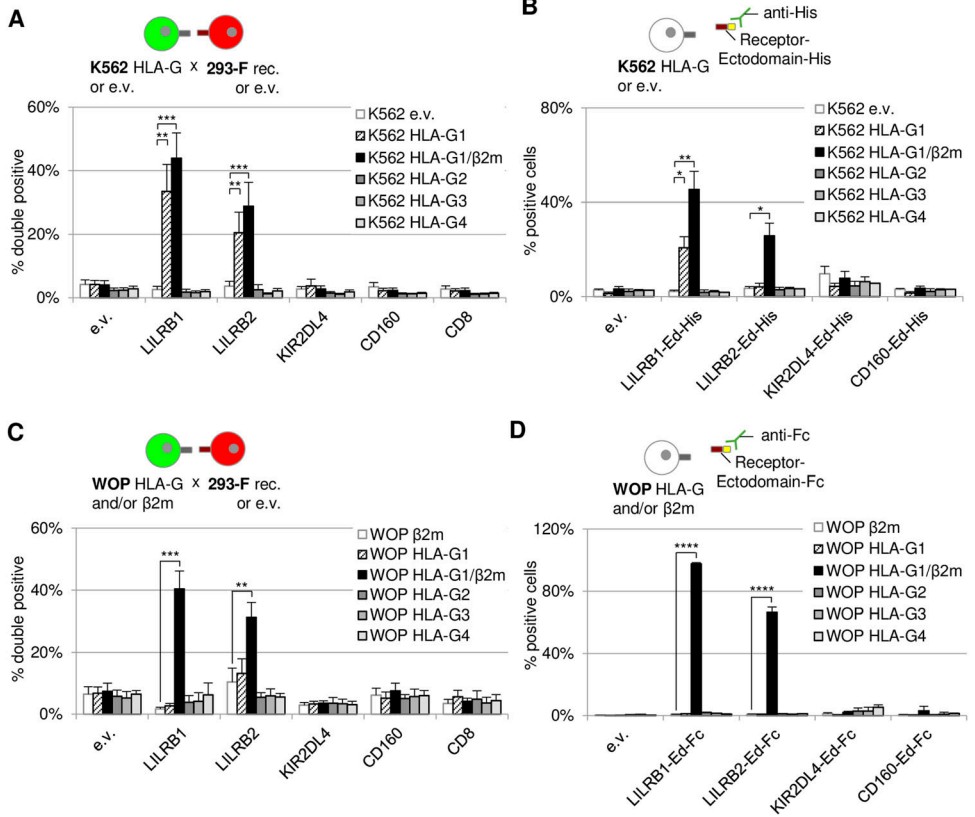

**Figure 5. Characterization of the HLA-G membrane isoform interaction network using Cell-Int.**
**(A, C)** Bar graph showing the percentage of double-positive events among RFP-positive cells, indicating cell interaction between receptor- and RFP-expressing 293-F cells and (A) K562 cells stably expressing GFP and HLA-G isoforms or (C) WOP cells stably expressing HLA-G isoforms and transfected by the GFP expression vector. **(B, D)** Bar graph showing the binding of (B) soluble His-tagged receptor ectodomains (Ed) to K562 or (D) soluble Fc-tagged receptor ectodomains to WOP cells stably expressing HLA-G isoforms, as assessed by the percentage of (B) anti-His or (D) anti-Fc–positive cells. **(A, B, C, D)** White bars indicate background binding to (A, B) e.v.-transduced K562 cells or (C, D) $\beta$2m-transduced WOP cells; hatched bars indicate binding to cells expressing HLA-G1 only; gray shades indicate binding to the different HLA-G isoform–expressing cells, from HLA-G1/$\beta$2m (black) to HLA-G4 (light gray). **(A, B, C, D)** Data show the mean ± SEM of at least three independent experiments. Statistics were performed using a paired $t$ test (*$P \le 0.05$; **$P \le 0.01$; ***$P \le 0.0001$; ****$P \le 0.0001$).

assay and therefore show that Cell-Int is a powerful and relevant technique for studying membrane protein interactions.

To further characterize the number of binding proteins at the cell surface that are required for the interaction between two cell populations to be detected using Cell-Int, we employed the model interaction between HLA-G1/$\beta$2m-expressing WOP cells and LILRB1-expressing 293-F cells (Fig S6A). The number of LILRB1 molecules expressed at the cell surface was estimated at various dilutions, ranging from 1 to 1/10,000[th], using the Quantibrite quantification method (Serke et al, 1998). This method allows the quantification of membrane proteins with PE-coupled antibodies. The corresponding interaction signals were assessed in parallel using Cell-Int. The interaction signal between LILRB1 and HLA-G1/$\beta$2m was observed to be increased in comparison with the e.v. control condition at dilutions of 1 to 1/300[th] of the LILRB1 expression vector, although the differences were only statistically significant at dilutions of 1 and 1/10[th] (Fig S6A, left panel). It is noteworthy that this interaction signal appears to correlate better with the percentage of LILRB1-expressing cells than with the number of surface LILRB1 proteins (Fig S6A, right panel). The correlation with the percentage of expressing cells was also evident when a second interacting pair was examined using the same methodology. The PD-1/PD-L1 interaction signal was significantly elevated at dilutions ranging from 1 to 1/300[th] of the PD-L1–expressing vector, and correlated with the percentage of PD-L1–expressing cells, rather than with the number of cell surface PD-L1 molecules (Fig S6B).

## The inhibitory activity of blocking antibodies can be assessed by Cell-Int

The therapeutic efficiency of several antibodies used in immunotherapy relies on the inhibition of membrane PPIs involved in immune down-regulation, such as anti-PD-1, anti-PD-L1, and anti-CTLA-4 antibodies, or in cell proliferation, such as anti-EGFR antibodies. As we have shown in Fig 2 that, using Cell-Int, we can detect the interaction between PD-1 and its ligands PD-L1 and PD-L2, as well as between EGFR and EGF fused to MeV H, we tested the inhibitory activity of increasing doses of three immunotherapeutic agents: nivolumab (anti-PD-1), atezolizumab (anti-PD-L1), and cetuximab (anti-EGFR) (Fig 6). The graphs in Fig 6A (non-normalized data) and Fig S7 (normalized data) show the nivolumab-mediated inhibition of PD-1 binding to its ligands PD-L1 and PD-L2, with IC$_{50}$ values of 1.18 and 0.99 $\mu$g/ml, respectively. As expected, the interaction between CTLA-4 and CD80 is not affected by the treatment. The cell interaction assay also detects the inhibition of PD-1/PD-L1 interaction by the anti-PD-L1 atezolizumab with an IC$_{50}$ of 1.62 $\mu$g/ml (Fig 6B). No effect on the PD-1/PD-L2 interaction signal is observed. Moreover, our results confirm the inhibitory effect of cetuximab on H-EGF/EGFR interaction with an IC$_{50}$ of 1.81 $\mu$g/ml (Fig 6C). As anticipated, cetuximab does not exert an inhibitory effect on the PD-1/PD-L1 interaction.

New immune checkpoints such as HLA-G and its receptors emerge as targets for cancer treatment (Amiot et al, 2011; Carosella et al, 2015; Barkal et al, 2018; Krijsman et al, 2020). Blocking HLA-G

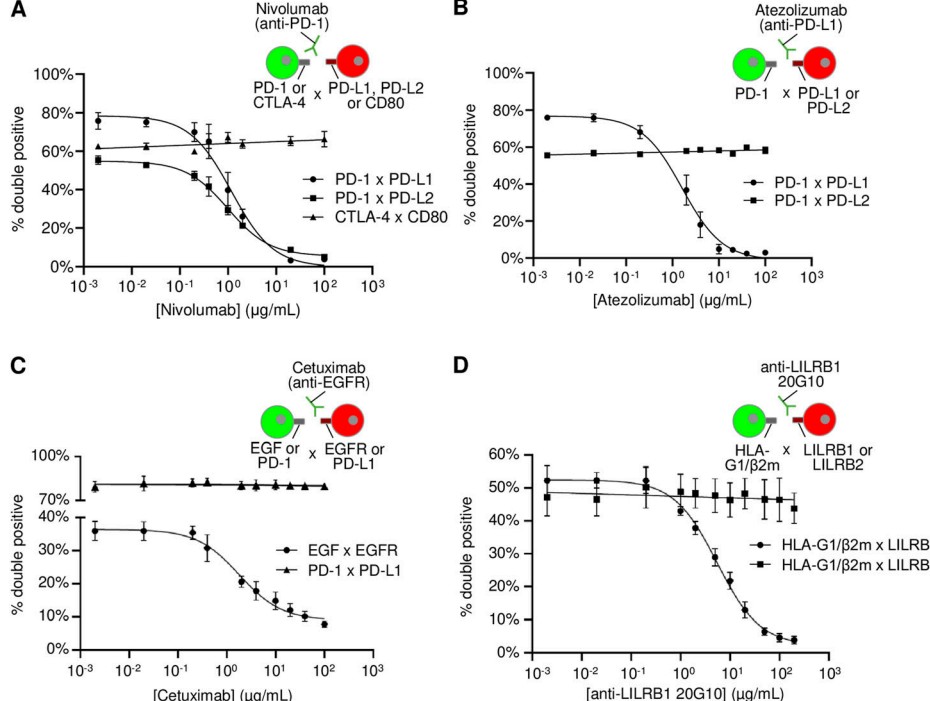

**Figure 6. Cell-Int detects the inhibitory activity of blocking antibodies.**

**(A, B, C, D)** Inhibitory activity of each blocking antibody was assessed by measuring cell interaction signal after pre-incubation of cells expressing the antibody-targeted membrane protein with different doses of antibody for 45 min at 25°C. Inhibitory dose–response curves show the percentage of double-positive events among RFP-positive cells, in the presence of increasing antibody concentrations (three to seven independent experiments). **(A)** Anti-PD-1 antibody (nivolumab) inhibitory activity was tested on PD-1/PD-L1 (●) and PD-1/PD-L2 (■) binding pairs, as well as CTLA-4/CD80 (▲) as a control. **(B)** Anti-PD-L1 antibody (atezolizumab) inhibitory activity was tested on PD-1/PD-L1 pair (●) and PD-1/PD-L2 (■) as a control. **(C)** Anti-EGFR antibody (cetuximab) inhibitory activity was tested on EGF/EGFR pair (●), as well as PD-1/PD-L1 (▲) as a control. **(D)** Anti-LILRB1 (20G10) inhibitory activity was tested on HLA-G1 $\beta$2m/LILRB1 pair (●) and HLA-G1 $\beta$2m/LILRB2 (■) as a control. **(A, B, C, D)** Cell interaction signals were measured (A, B, C) between GFP and RFP 293-F cells co-transfected with indicated membrane proteins, or (D) between WOP cells stably expressing HLA-G1/$\beta$2m and GFP, and RFP 293-F cells co-transfected with the LILRB1 or LILRB2 expression vector. Data were analyzed using GraphPad Prism.

receptors, such as LILRB1, could lead to the inhibition of HLA-G immune-suppressive signaling. We used a newly identified anti-LILRB1 antibody and showed that it successfully inhibits the interaction between HLA-G1/$\beta$2m and LILRB1 using the cell interaction assay, with an $IC_{50}$ value of 5.47 $\mu$g/ml (Fig 6D). As anticipated, the interaction between HLA-G1/$\beta$2m and LILRB2 was not found to be inhibited by this antibody. In view of these results, Cell-Int is a suitable technique for evaluating the inhibitory potential of new blocking antibodies in quasi-physiological conditions.

## Discussion

Identifying and characterizing intercellular membrane PPIs still poses technical challenges. Here, we have described the development of a novel versatile technique, designated Cell-Int, that uses live cells to study membrane PPIs within the native, physiologically relevant plasma membrane environment. Because the proteins are expressed by mammalian cells, they should exhibit appropriate post-translational modifications, including glycosylations and disulfide bonds. Unlike many available methods to characterize extracellular interactions, we have shown that most types of surface proteins can be studied using Cell-Int, including type I (e.g., PD-1, CTLA-4, SARS-CoV-2 Spike, EGFR…) and II (COLEC12, MeV H) single-pass transmembrane proteins, GPI-anchored

proteins (Juno, Ly-6E), multipass membrane proteins (MFSD2, Slc19A1), and multimeric structures such as HLA-G1/$\beta$2m dimer, or even trimers of dimers such as Syncytins.

Engineering a fusion protein combining portions of two or more proteins is a sensible strategy for incorporating novel binding specificities into a surface protein. Such a strategy is used for viruses serving as vectors for gene delivery and for oncolytic viruses to modify their tropism by incorporating a ligand onto viral surface glycoproteins (Lavillette et al, 2001; Navaratnarajah et al, 2012; Goins et al, 2016). We have shown, by validating the ability of MeV H glycoprotein fused to EGF to significantly bind EGFR, that Cell-Int is also a simple relevant tool to check the efficacy of a retargeting strategy. Similarly, we have shown that our technique is also relevant for evaluating the CAR-mediated targeting of cancer cells, which are rapidly expanding anticancer therapeutics (Sterner & Sterner, 2021).

We have demonstrated that our technique is sensitive enough to detect intercellular PPIs with affinities in the $\mu$M-to-nM range (Table 1), with Juno/Izumo1 and PD-1/PD-L1 complexes displaying the weakest interactions in terms of affinity. The HLA-G1/CD8 Kd value of 72 $\mu$M represents a potential sensitivity threshold for the Cell-Int technique, as no interaction signal was observed for this pair. The technique's ability to detect low-affinity cell surface interactions is probably facilitated by an increase in binding avidity because of the local concentration of overexpressed proteins at cell–cell contacts. However, the interaction signals measured by Cell-Int do not correlate with affinities measured

**Table 1.** Affinity constants of receptor/ligand complexes tested in this study.

| Complex | Affinity ($K_d$) | Technique | Reference |
|---|---|---|---|
| Juno/Izumo1 | 12.3 ± 0.2 μM | SPR | Bianchi et al (2014) |
| PD-1/PD-L1 | 8.2 ± 0.1 μM | SPR | Cheng et al (2013) |
| PD-1/PD-L2 | 2.3 ± 0.1 μM | SPR | Cheng et al (2013) |
| CD80/CTLA-4 | 0.42 ± 0.06 μM | SPR | van der Merwe et al (1997) |
| CD200/CD200R1 | 0.49 ± 0.08 μM | SPR | Wright et al (2003) |
| PILRα/COLEC12 | 1.1 μM | SPR | Sun et al (2012) |
| PILRα/NPDC-1 | 49 nM | Equilibrium competition radioligand assay | Sun et al (2012) |
| SARS-CoV-2 Spike RBD/ACE2 | 44.2 nM | SPR | Shang et al (2020) |
| MeV H/CD46 | 79 nM | SPR | Navaratnarajah et al (2008) |
| MeV H/SLAM | 80 nM | SPR | Navaratnarajah et al (2008) |
| MeV H/Nectin-4 | 20.1 nM | SPR | Mühlebach et al (2011) |
| Syncytin-2/MFSD2 | ND | N/A | N/A |
| Syncytin-A/Ly-6E | ND | N/A | N/A |
| GLN Env/Slc19A1 | ND | N/A | N/A |
| H-EGF/EGFR | ND | N/A | N/A |
| HLA-G1/LILRB1 | 2.0 ± 0.7 μM | SPR | Shiroishi et al (2003) |
| HLA-G1/LILRB2 | 4.8 ± 1.4 μM | SPR | Shiroishi et al (2003) |
| HLA-G1/KIR2DL4 | ND | N/A | N/A |
| HLA-G1/CD160 | ND | N/A | N/A |
| HLA-G1/CD8a | 72.0 ± 1.4 μM | SPR | Shiroishi et al (2003) |

ND, not determined; N/A, not applicable.

using conventional techniques such as surface plasmon resonance (data not shown). It is likely that this absence of correlation is attributable, at least in part, to the potential conformational differences between soluble ectodomains, which are used in most techniques for assessing affinities, and full membrane-embedded proteins, which are studied using Cell-Int.

Interestingly, our results illustrate that despite very similar extracellular ligand-binding regions, immunoregulatory paired receptors do not necessarily interact with the same ligands. In particular, our results suggest that the activating receptor PILRβ, which shares 87% similarity to its inhibitory counterpart PILRα in their extracellular amino acid sequences, does not interact with two previously described interaction partners of PILRα, namely, NPDC-1 and COLEC12 (Sun et al, 2012). Similarly, our results confirm that the activating receptor CD200R1L does not interact with CD200, unlike the inhibitory receptor CD200R1 (Wright et al, 2003; Pascoal Ramos et al, 2022). A review of the literature on paired receptors by Kuroki and colleagues (Kuroki et al, 2012) reveals that inhibitory receptors have been demonstrated to bind to a greater number of ligands. In the case of PILRs, the PILRα and PILRβ structures were compared in the study by Lu et al (2014). Using the example of HSV-1 glycoprotein B (gB)/PILRα interaction, they demonstrate that the mutation of a single PILRβ residue restores the interaction with gB. It would be interesting to conduct similar structural studies on the PILRα/NPDC-1 interaction to determine whether it is also possible to identify the PILRα residues involved in the interaction and why this interaction is not detected with PILRβ.

Antibody-mediated antitumor therapies with the example of checkpoint inhibitors have revolutionized cancer treatment. As new therapeutic antibodies are constantly being developed, aimed at blocking oncogenic cell signaling by inhibiting extracellular PPIs (Tsao et al, 2021; Jin et al, 2022), techniques to characterize their inhibitory effect are essential. We demonstrated here that Cell-Int is also relevant for assessing the inhibitory potential of therapeutic antibodies, as illustrated by the testing of three clinically approved monoclonal antibodies: nivolumab, atezolizumab, and cetuximab. The $IC_{50}$ values determined by Cell-Int are in a similar range to those reported by previous studies using different techniques (Table S1) (Wang et al, 2014; Bovio et al, 2020; Li et al, 2021), thereby demonstrating the relevance of Cell-Int in this context. HLA-G and its receptors are considered as a new immune checkpoint and potential targets for cancer immunotherapy (Amiot et al, 2011; Carosella et al, 2015; Barkal et al, 2018; Krijgsman et al, 2020), and several antibodies targeting HLA-G, LILRB1, or LILRB2 are currently in clinical trials for cancer treatment (Lin & Yan, 2021). Importantly, we validated in this study the inhibitory activity of a newly identified anti-LILRB1 antibody, which opens up perspectives for its use in cancer therapies. This result further shows that Cell-Int is a straightforward and effective tool for testing new blocking antibodies in quasi-physiological settings.

Immune evasion is not only a hallmark of cancer cells but has also been described for many microorganisms. Several types of pathogens, including viruses (e.g., human cytomegalovirus UL18 protein), bacteria, and unicellular protozoan parasites (e.g.,

**Table 2. Techniques to assess interaction/proximity between membrane proteins.**

| | | Parameter: | Considerations on microenvironment, protein expression, and conformation — Mb expr. | Endo. expr. | PTM | Intact prot. | All TM prot. | Practical considerations — Meas. param. | Ease to use, cost, and other disadvantages | References |
|---|---|---|---|---|---|---|---|---|---|---|
| Cellular approaches | Proximity labeling methods based on biotinylation | APEX/HRP | Yes | Yes | Yes | No (bait prot. are fused) | Yes | Proximity | Requires genetic constructs; $H_2O_2$ oxidizing capacity may affect protein folding; only ligands with tyrosines can be biotinylated. | Rhee et al (2013) and Loh et al (2016) |
| | | BioID/TurboID | | | | No (bait prot. are fused) | | | Requires genetic constructs; only ligands with free amines can be biotinylated. | Roux et al (2012) and Takano et al (2020) |
| | | µMap | | | | Yes | | | Requires the use of an antibody against the protein. | Geri et al (2020) |
| | | split HRP PCA | | | | No (prot. are fused) | | | Requires genetic constructs; HRP folding requires heme; $H_2O_2$ oxidizing capacity may affect protein folding; only ligands with free amines can be biotinylated. | Martell et al (2016) |
| | | uLIPSTIC | | | | Yes | | | Reveals cell interactome rather than protein interactome; requires genetic constructs; the large number of cells is required. | Nakandakari-Higa et al (2024) |
| | Other split systems | SIMPL | Possible | No | No | No (prot. are fused) | | | Requires genetic constructs. | Yao et al (2020) |
| | | Mb yeast 2-hybrid | Yes | | No | No (prot. are fused) | | | Requires genetic constructs; partners are expressed in the same yeast cell, only cis interactions are detected. | Thaminy et al (2003) |
| | Cell-based assays using cDNA libraries | Extracell. bind. assays/ Retrogenix microarray | Only for the rec. | | Yes | No (bait prot. are fused) | Mainly single pass | | Requires genetic constructs and protein production. | Nakayama et al (1992) and Turner et al (2013) |
| | | Cell-Int | Yes | No (but possible) | | Yes | Yes | | Potential sensitivity limitations for very weak interactions. | This study |
| Soluble approaches | ELISA-like, binding to soluble/anchored receptor | AVEXIS | No | No | No | No, (Ed, oligom. ligand, tagged rec.) | Mainly single pass | Interaction | Requires genetic constructs and protein production. | Bushell et al (2008) and Martinez-Martin et al (2018) |
| | | RDIMIS | Rec. in EV mb | | Yes | No (ligands are Fc-tagged Ed) | | | Requires genetic constructs and protein production; needs to isolate EV from producer cells. | Cao et al (2021) |
| | | Mb protein array, NAPPA array | Possible (microsomal mb) | | No (but possible) | No (prot. are tagged) | | | Requires genetic constructs; cell-free system makes protein folding uncertain; washes make it difficult to study weak interactions. | Ramachandran et al (2008), Ramani et al (2012), and Glick et al (2016) |
| | Ligand capture | TRICEPS | Rec. can be | For the rec. | Yes | No (ligands are tagged) | Yes | | Ligand is modified at its amine residues, and functionality may be altered; identifies N-glycosylated proteins only; oxidation may impact interaction. | Frei et al (2013) |
| | | AP-MS | Possible (requires cell lysis) | Yes, possible | | In some cases, prot. are tagged | | | In some cases, it requires an antibody against the protein; washes make it difficult to study weak interactions. | Satoh et al (2008) |

Most commonly used techniques for screening for interacting and/or proximal proteins are listed, including a summary of their characteristics and a list of references. Mb, membrane; expr., expression; endo., endogenous; PTM, post-translational modifications; prot., protein; TM, transmembrane; meas. param., measured parameter; HRP, horseradish peroxidase; PCA, protein-fragment complementation assay; extracell., extracellular; bind., binding; AP-MS, affinity purification–mass spectrometry; rec., receptor; EV, extracellular vesicles; Ed, ectodomains; oligom. ligand, oligomerized ligand.

Plasmodium falciparum RIFINs), have evolved to subvert the host immune system by targeting inhibitory receptors such as LILRBs (Cosman et al, 1997; Saito et al, 2017; Harrison et al, 2020; Abdallah et al, 2021; Sakoguchi et al, 2021). Using SARS-CoV-2 Spike, measles virus H protein, or endogenous retroviral proteins such as Syncytins as examples, we have shown that our technique is relevant for the assessment of virus–receptor interactions. It could be extended to the study of other host–pathogen interactions that involve plasma

membrane proteins. Our technique could be used to further characterize pathogen strategies involved in immune evasion by identifying novel host–pathogen interactions through library screening.

The current methods for identifying new membrane protein interactors, as outlined in Table 2, include mass spectrometry–based techniques such as affinity purification–mass spectrometry (Satoh et al, 2008) or TRICEPS (Frei et al, 2013). However, these approaches are dependent on the level of endogenous expression and may fail to identify receptors that are weakly expressed. In ELISA-like in vitro screens, such as AVEXIS (Bushell et al, 2008) and RDIMIS (Cao et al, 2021), the ligand is expressed either as a pentamerized soluble form or in the membrane of extracellular vesicles. Its binding to a soluble Fc-tagged immobilized receptor is then assessed. Consequently, neither or only one of the partners is in its native membrane environment. The development of proximity labeling methods based on biotinylation (Table 2), such as APEX/HRP-using techniques (Rhee et al, 2013; Loh et al, 2016), BioID and related techniques (Roux et al, 2012; Takano et al, 2020), or μMap approach (Geri et al, 2020), enables the direct study of proteins of interest in the plasma membrane or even in specific subdomains of the membrane. However, these techniques rely on labeling proteins present in the microenvironment rather than directly interacting with the ligand of interest. Consequently, although these methods permit the identification of proteins that are enriched in a specific microenvironment, a direct interaction would necessitate further characterization through alternative approaches. In addition, some of these methods rely on the use of cofactors, such as $H_2O_2$, which are inherent to HRP-derived techniques and have the potential to affect protein folding. Furthermore, most of these methods require the use of numerous genetic constructs, which renders them both time-consuming and costly. To address the need for alternative large-scale screening techniques capable of detecting weak interactions between membrane proteins in quasi-physiological conditions, and unhampered by endogenous expression levels, we took advantage of our cell interaction assay to develop a new, efficient screening method. In particular, we performed an unbiased screen for HLA-G1 extracellular interaction partners using a cDNA library spanning ~12,000 human genes. We confirmed that the previously described LILRB1 receptor induces cell-to-cell interaction with HLA-G1–expressing cells. We further tested the interaction of previously described receptors with the four HLA-G1–4 membrane isoforms to find that only the previously described LILRB1 and LILRB2 receptors can induce cell-to-cell interaction by binding HLA-G1, whereas three other receptors cannot.

Importantly, we found that β2m expression was required for HLA-G1 to interact with both LILRB1 and LILRB2 in our experiments. Gonen-Gross and colleagues showed that HLA-G1 free heavy chains could not bind to LILRB1 (Gonen-Gross et al, 2005), and the importance of the β2m for the interaction with this receptor was reported in different studies (Shiroishi et al, 2006; HoWangYin et al, 2012). The same two studies also reported that LILRB2 could bind HLA-G1 in the absence of β2m, which is in apparent contradiction with our results. In those studies, however, the importance of β2m was investigated under non-physiological conditions, mainly using soluble forms of HLA-G1 or HLA-G5. The study conducted by Arns and colleagues (Arns et al, 2020) showed

using structural modeling that the residues involved in the interaction between HLA-G and LILRB2 (and LILRB1) are less accessible in the membrane-bound HLA-G1 than in the soluble form. We can hypothesize that in the plasma membrane environment, by increasing the stability of HLA-G1, as suggested by the aforementioned study (Arns et al, 2020), and its expression at the cell surface as observed in Fig S5B, β2m is required for HLA-G1 to interact with LILRB2.

Among the other HLA-G receptors investigated in the present work, KIR2DL4 has given rise to divergent results in the literature, with some experiments reporting the absence of interaction at the cell membrane (Allan et al, 1999) or through surface plasmon resonance (Boyson et al, 2002), whereas others have suggested the interaction to be intracellular (Rajagopalan et al, 2006). Indeed, soluble HLA-G was found to be internalized in endosomes in a process involving KIR2DL4 (Rajagopalan et al, 2006). Although KIR2DL4 might reside in the endocytic compartment in resting NK cells, it is well addressed to the plasma membrane in our cell lines, and yet, we could not detect any interaction with membrane-bound HLA-G1. We cannot exclude, however, that a cofactor, absent in our experiment, is involved in HLA-G binding or that the endocytic environment is required for the interaction to occur. CD8 is a common receptor for MHC class I proteins, with two studies showing that it binds soluble HLA-G1 (Contini et al, 2003; Shiroishi et al, 2003). However, the affinity of HLA-G1/CD8 interaction is much lower than that of HLA-G1/LILRBs (Kd = 72 μM for CD8; 2 and 4.8 μM for LILRB1 and LILRB2, respectively) (Shiroishi et al, 2003). The weak binding measured in vitro might explain the absence of interaction detected by the cell–cell interaction assay in a physiologically relevant context. Regarding CD160, this receptor is poorly described in the literature. The protein was identified as interacting with HLA-G1 with a supposedly weak affinity (Agrawal et al, 1999) leading to inhibition of angiogenesis (Fons et al, 2006). However, no other study has since confirmed the interaction. Our results suggest that under quasi-physiological conditions, CD160 does not interact significantly with HLA-G1.

In the present study, we also investigated the interaction of the three other membrane-bound isoforms HLA-G2–4. To our knowledge, this is the first study to provide a comprehensive overview of HLA-G membrane isoform interaction network with previously described receptors. None of the five receptors interacted with any of the other membrane-bound HLA-G isoforms. The latter are largely uncharacterized in the literature, especially HLA-G3 and HLA-G4. Regarding HLA-G2, some authors have reported its benefits when using it as a therapeutic soluble molecule (HoWangYin et al, 2012; LeMaoult et al, 2013; Takahashi et al, 2016). These studies suggested that LILRB2 can also act as a receptor for HLA-G2. In both studies, however, the biochemical techniques documenting the interaction, such as Bio-Plexing and immunoprecipitation, are performed under non-physiological conditions, mainly on soluble protein ectodomains, which may explain the discrepancy with our data.

Overall, Cell-Int can provide a multipurpose tool for studying most extracellular PPIs in quasi-physiological conditions. We believe it will contribute to a comprehensive understanding of the membrane interactome and how to modulate it for

**Table 3.  PCR primers used in this study for cloning.**

| Primer | | Sequence (5′–3′) |
|---|---|---|
| Izumo1 | fwd | ATACATCTCGAGGCCACCATGGGGCCGCATTTTA |
| | rev | ATACATACGCGTTTAGTTTTCTGTTGCCTCGCT |
| PD-1 | fwd | ATACATCTCGAGGCCACCATGCAGATCCCACAGGC |
| | rev | ATACATACGCGTTCAGAGGGGCCAAGAGCA |
| PD-L1 | fwd | ATACATCTCGAGGCCACCATGAGGATATTTGCTGT |
| | rev | ATACATACGCGTTTACGTCTCCTCCAAATG |
| PD-L2 | fwd | ATACATCTCGAGGCCACCATGATCTTCCTCCTGCT |
| | rev | ATACATACGCGTTCAGATAGCACTGTTCAC |
| CD80 | fwd | ATACATCTCGAGGCCACCATGGGCCACACACGGAG |
| | rev | ATACATACGCGTTTATACAGGGCGTACACT |
| CD200 | fwd | ATACATCTCGAGGCCACCATGGAGAGGCTGGTGAT |
| | rev | ATACATACGCGTTTAGGGCTCTCGGTCCTG |
| CD200R1 | fwd | ATACATCTCGAGGCCACCATGCTCTGCCCTTGGAG |
| | rev | ATACATACGCGTTTATAAAGTATGGAGGTC |
| CD200R1L | fwd | ATACATCTCGAGGCCACCATGTCAGCTCCAAGATT |
| | rev | ATACATACGCGTCTAAAGAACTTTTCTGAC |
| PILRα | fwd | ATCATTTCCGGAAGCCACCATGGGTCGGCCCCTG |
| | rev | GTCCTTCTAGACCAGGCCTTTAAGACAG |
| PILRβ | fwd | ATCATTTCCGGACACCGGTGCCACCATGGGTCGGCCCCTG |
| | rev | GTCCTTCTAGACGCGGCCGCGAAGTCACTGCTTG |
| COLEC12 | fwd | AGCATTTCCGGAAGCCACCATGAAAGACGACTTC |
| | rev | GGCGTTCTAGACCATAATGCAGATGACAG |
| NPDC1 | fwd | ATACATTCCGGAAGGATGGCGACGCC |
| | rev | ATACATTCTAGACCATGGCAGTGCAGGCG |
| Nectin-4 | fwd | ATACATCTCGAGGCCACCATGCCCCTGTCCCTGGG |
| | rev | ATACATACGCGTTCAGACCAGGTGTCCCCG |
| CD46 | fwd | TTGGGTGATCACTCGAGCGCGCATGGAGCCTCCC |
| | rev | ATACATGCGGCCGCACGCGTGGTTGTGGAATCTATTCAGC |
| SLAM | fwd | ATACATTGATCACTCGAGGCCACCATGGATCCCAAGGGGC |
| | rev | ATACATGCGGCCGCACGCGTTCAGCTCTCTGGAAGTGT |
| HLA-G1/-G2 phCMV | fwd | ATACATCTCGAGGCCACCATGGTGGTCATG |
| | rev | ATACATACGCGTTCAATCTGAGCTCTTCTTTCTCCA |
| HLA-G3 | fwd | ATACATCTCGAGGCCACCATGGTGGTCATG |
| | rev | AGGGAAGACTGCTTCGCCTCGCTCTGGTTGTAGTAGCCG |
| HLA-G4 | fwd | ATACATCTCGAGGCCACCATGGTGGTCATG |
| | rev | AGGGAAGACTGCTTCGCGCGCTGCAGCATCTCCTTCCC |
| HLA-G TM | fwd | AAGCAGTCTTCCCTGCCCAC |
| | rev | ATACATACGCGTTCAATCTGAGCTCTTCTTTCTCCA |
| HLA-G lentiviral vector | fwd | ATACATGGATCCGCCACCATGGTGGTCATGGCGCCCCG |
| | rev | ATACATGCGGCCGCTCAATCTGAGCTCTTCTTTCTC |
| β2m lentiviral vector | fwd | ATACATGGATCCGCCACCATGTCTCGCTCCGTGGCCTT |
| | rev | ATACATGCGGCCGCTTACATGTCTCGATCCCACT |

**Table 3.   Continued**

| Primer | | Sequence (5′–3′) |
|---|---|---|
| LILRB1 phCMV | fwd | ATACATCTCGAGATGACCCCCATCCTCACGGTCC |
| | rev | ATACATACGCGTCTAGTGGATGGCCAGAGTGG |
| LILRB2 phCMV | fwd | ATACATCTCGAGATGACCCCCATCGTCACAGT |
| | rev | ATACATACGCGTTTAGTGGATGGCCAGGGTG |
| KIR2DL4 phCMV | fwd | ATTGTACTCGAGATGTCCATGTCACCCACGGT |
| | rev | ATACATACGCGTTCAGATTCCAGCTGCTGGTA |
| CD160 phCMV | fwd | ATTGTACTCGAGATGCTGTTGGAACCCGGC |
| | rev | ATACATACGCGTTTACAAAGCTTGAAGGGCCA |
| CD8a phCMV | fwd | ATTGTACTCGAGATGGCCTTACCAGTGACCGC |
| | rev | ATTGTAACGCGTTTAGACGTATCTCGCCGAAA |
| CD8b phCMV | fwd | ATTGTACTCGAGATGCGGCCGCGGCTGTGG |
| | rev | CTCCATACGCGTCTACTTGTAGAACTGTTTCATGAAACGAAG |
| LILRB1 Fc | fwd | ATACATGATATCGCACCTCCCCAAGCCCACCCTCT |
| | rev | ATACATGCGGCCGCGTGCCTTCCCAGACCACTCT |
| LILRB2 Fc | fwd | ATACATGATATCGGGGACCATCCCCAAGCCCACCC |
| | rev | ATACATGCGGCCGCCAGGTGCCTTCCCAGACCAC |
| KIR2DL4 Fc | fwd | ATACATGATATCGCACGTGGGTGGTCAGGACAAGCC |
| | rev | ATACATGCGGCCGCGTGTCTGGCGATACCAGTTT |
| CD160 Fc | fwd | ATACATGATATCGGGATGCATTAACATCACCAGCT |
| | rev | ATACATGCGGCCGCACTGAGAGTGCCTTCATTAT |
| LILRB1 His | fwd | ATACATGCGGCCGCCACCATGACCCCCATCCTCACG |
| | rev | ATACATTTCGAAGTGCCTTCCCAGACCACTCTGGG |
| LILRB2 His | fwd | ATACATGCGGCCGCCACCATGACCCCCATCGTCACA |
| | rev | ATACATTTCGAACAGGTGCCTTCCCAGACCACT |
| KIR2DL4 His | fwd | ATACATGCGGCCGCCACCATGTCCATGTCACCCACG |
| | rev | ATACATTTCGAAGTGTCTGGCGATACCAGTTTTG |
| CD160 His | fwd | ATACATGCGGCCGCCACCATGCTGTTGGAACCCGGC |
| | rev | ATACATTTCGAAACTGAGAGTGCCTTCATTATGGC |

therapeutic purposes by incorporating new ligands onto surface proteins or by using inhibitory antibodies.

# Materials and Methods

### Cell lines

Human 293T embryonic kidney cells and murine WOP embryonic fibroblast cells were cultured at 37°C and 5% $CO_2$ in DMEM (Thermo Fisher Scientific) supplemented with 10% heat-inactivated FBS (Thermo Fisher Scientific), 100 µg/ml streptomycin (Thermo Fisher Scientific), and 100 U/ml penicillin (Thermo Fisher Scientific). Human FreeStyle 293-F cells (R79007; Thermo Fisher Scientific) were cultured at 37°C, 8% $CO_2$, and 120 rpm (Celltron shaker, Infors HT, throw 25 mm) in FreeStyle 293 Expression Medium (Thermo Fisher Scientific). Human Raji and K562 lymphoblast cells were cultured at 37°C and 5% $CO_2$ in Roswell Park Memorial Institute (RPMI) 1640 medium (Thermo Fisher Scientific) supplemented with 10% heat-inactivated FBS, 100 µg/ml streptomycin, and 100 U/ml penicillin. Cells were tested negative for mycoplasma contamination.

### Expression vectors

Several expression vectors used in this study were derived from phCMV-VSV-G (GenBank accession number AJ318514; a gift from F.-L. Cosset). This includes the empty vector (e.v.) control and the previously described vectors for the expression of human Syncytin-2 (Blaise et al, 2003), mouse Syncytin-A (Dupressoir et al, 2005), and GLN Env (Ribet et al, 2008). The cDNAs encoding human PD-1, PD-L1, PD-L2, CD80, CD200, CD200R1, CD200R1L, Nectin-4, and SLAM were subcloned from the ORFeome cDNA library (see the section on cDNA library screening and subcloning) into the phCMV-derived vector using an XhoI-containing forward primer with a Kozak

sequence before the initiation codon and a MluI-containing reverse primer specific to the C-terminal sequence (all primers are listed in Table 3). *CD46* was cloned from human placenta–derived cDNA also into a phCMV-derived vector using similar primers (Table 3). The cDNA encoding mouse protein Izumo1 was subcloned from a pCAGGS-Izumo1-IRES-RFP vector (a gift from C. Gourier) (Chalbi et al, 2014) into the phCMV-derived vector similarly (Table 3).

*NPDC-1* was cloned from human PBMC-derived cDNA into the pCMV4-3xHA vector using a BspEI-containing forward primer and an XbaI-containing reverse primer (Table 3). *COLEC12* and *PILRβ* cDNAs were subcloned from the ORFeome cDNA library (see below) also into the pCMV4-3xHA vector with similar primers. *PILRα* cDNA was subcloned from the plasmid RDC1528 (R&D Systems) into the pCMV4-3xHA vector similarly.

The CTLA-4 expression vector (pCMV3-CTLA-4; HG11159-UT) was purchased from Sino Biological. EGFR (pHAGE-EGFR; 116731; a gift from G. Mills and K. Scott) (Ng et al, 2018) and ACE2 expression vectors (pcDNA3.1-hACE2; 145033; a gift from F. Li) (Shang et al, 2020) were from Addgene. The peGFP-C3 (BD Biosciences) and pCMV6-AC-RFP (CliniSciences) plasmids were used for expressing GFP and RFP, respectively. The vector for the expression of mouse protein Juno (a gift from G. J. Wright) (Bianchi et al, 2014), MFSD2 (Esnault et al, 2008), mouse Ly-6E (Bacquin et al, 2017), mouse Slc19A1 (Tsang et al, 2019), SARS-CoV-2 Spike (CoV-2 Spike) (pCG1-SARS-2-S; a gift from S. Pöhlmann) (Pfaender et al, 2020), measles virus (MeV) H protein, and H-EGF fusion protein (pCG-H and pCG-H-αEGFR, respectively; a gift from C. J. Buchholz) (Cathomen et al, 1995; Funke et al, 2008) were previously described. The ErbB-2 expression vector was isolated from the ORFeome library (see below).

Anti-CD19 and anti-EGFR second-generation CAR lentiviral vectors expressing respectively anti-CD19 ScFv (Fmc63) and anti-EGFR ScFv from nimotuzumab, along with GFP via an IRES sequence, were described previously (Marchais et al, 2023).

*HLA-G1* cDNA was subcloned from pCMV3-HLA-G1 (Sino Biological; HG17351-UT). The β2-microglobulin coding sequence was cloned in phCMV from the pcDNA3.1 ZEO(+) hB2M vector (a gift from J. Le Maoult). The HLA-G2 coding sequence was constructed by digesting *HLA-G1* cDNA with AlwNI and SbfI and inserting the α1–α3 junction of the HLA-G6 sequence (a gift from J. Le Maoult) digested by the same enzymes. HLA-G3 and HLA-G4 were constructed by multifragment PCR allowing the junction between α1 or α2 domains with the transmembrane region of HLA-G1 using HLA-G3 or HLA-G4 with HLA-G TM primers (Table 3). For transient membrane expression, HLA-G isoform coding sequences were cloned into the phCMV vector through XhoI/MluI restriction sites. For lentiviral production, HLA-G isoforms or β2-microglobulin coding sequences were cloned through BamHI/NotI sites into a vector derived from a pDual lentiviral vector, where one of HLA-G membrane isoforms or β2-microglobulin (or nothing in the case of the empty vector control) is co-expressed with the hygromycin resistance gene (Perro et al, 2010). The GFP lentiviral vector was previously described (Tsang et al, 2019). Plasmid 8.91 was used to produce HIV-1 core proteins (Zufferey et al, 1997). *LILRB2* cDNA was subcloned from pGEM-LILRB2 (Sino Biological; HG14132-G). The other HLA-G receptor cDNAs (i.e., LILRB1, KIR2DL4, CD160, and CD8a and CD8b) were subcloned from the ORFeome library (see below; Table 3). For transient membrane expression, HLA-G receptor cDNAs were

cloned into the phCMV vector through XhoI/MluI restriction sites. For the production of soluble Fc-tagged proteins, cDNAs were cloned into the pFuse-hIgG1-Fc2 vector, downstream of the IL-2 signal sequence, through EcoRV/NotI sites for all constructs except for CD8 where NcoI/NotI sites were used. For the production of soluble His-tagged proteins, genes were cloned into the pCDNA3.1/V5-His-TOPO vector, with their own signal sequence, using NotI/BstBI sites.

## Anti-LILRB1 20G10 antibody generation and production

Monoclonal antibodies (mAbs) were raised in Biozzi mice by immunizing with LILRB1-Fc (30 µg per injection) fusion protein produced in CHO cells. Mice eliciting the highest anti-LILRB1-Fc antibody response were given an intravenous boost injection 3 d before being euthanized for splenic B-cell fusion. Hybridoma culture supernatants were screened for antibody production, specificity for LILRB1, and blocking capacity with MHC-I, by enzyme immunoassay, flow cytometry, or both. Selected hybridomas were subsequently cloned by limiting dilution. Monoclonal antibodies were produced in hybridoma supernatants and further purified using the AKTA protein purifier. The purity of mAbs was assessed by SDS–PAGE in reducing and non-reducing conditions using Agilent 2100 Bioanalyser. Endotoxin levels in purified mAbs were determined by Pierce LAL Chromogenic Endotoxin Quantitation Kit (Thermo Fisher Scientific) according to the manufacturer's protocol.

## Lentiviral transduction

Stable WOP and K562 cell lines expressing proteins of interest were obtained through lentiviral transduction. $2 \times 10^6$ 293T cells were seeded in 10-cm culture dishes. The next day, cells were transfected with plasmids encoding the VSV glycoprotein (phCMV-VSV-G), the lentiviral core genome (8.91), and the lentiviral transgene expression vector at a ratio of 1:2:3, using jetPRIME (Ozyme) transfection reagent and following the manufacturer's protocol. Target WOP cells were plated the day before infection at $5 \times 10^4$ cells per well in a six-well plate, whereas K562 cells were seeded just before infection at $10^6$ cells/ml in a 12-well plate. 3 d after transfection, lentiviral supernatants were collected, centrifuged, and filtered through a 0.45-µm filter. Infections were then performed by adding 1 ml of lentiviral supernatants to target cells in the presence of polybrene (8 µg/ml). Cells were then subjected to spinoculation (centrifugation for 2 h at 1,200*g*, 25°C). After 3 d of infection, cells were either sorted or selected with hygromycin treatment (200 µg/ml for 2 wk; Dutscher, 702642). For sorting, WOP cells were detached with 10 mM EDTA in PBS. WOP and K562 cells were washed in PBS supplemented with 2% FBS and incubated in fluorophore-conjugated antibodies (PE-conjugated anti-HLA-G clone MEM-G/9, Exbio, 1P-292-C100, and/or PE-conjugated anti-human β2-microglobulin, 395704; BioLegend) diluted at 1:200 in PBS/2% FBS, for 45 min at RT. After washing in PBS/2% FBS, cells were resuspended in their culture medium and positively stained cells were collected using either INFLUX or ARIA Fusion 2 cell sorter (BD Bioscience).

The production of CAR lentiviruses and subsequent infection of 293-F cells were done following the previously described methods (Marchais et al, 2023).

### Cell interaction assay (Cell-Int)

#### Cell assay between two populations of 293-F cells

24 h before transfection, 293-F cells were seeded in 6- or 12-well low-adhesion plates at $10^6$ cells per ml. 293-F cells were transiently transfected using FectoPro reagent (Polyplus) following the manufacturer's protocol. A first batch of cells was co-transfected with an equivalent DNA quantity of a vector encoding a membrane protein of interest (or an e.v. control) and a GFP expression vector. A second batch of cells was transfected with an equivalent DNA quantity of a vector encoding RFP and a second membrane protein of interest (or an e.v. control), except for the LILRB2 expression vector, which was diluted 10-fold to prevent LILRB2-induced cell mortality. The level of transfection was considered satisfactory if the percentage of GFP- and RFP-expressing cells was greater than 30% by cytometry analysis. 24 h post-transfection, GFP- and RFP-expressing cells were then mixed in equal volumes in their own culture medium (FreeStyle 293 Expression Medium), at a density of 1.5–3 × $10^6$ cells/ml. They were incubated for 1–2 h at 25°C (unless otherwise stated) until cytometry analysis. For the competition assay with inhibitory antibodies, before GFP and RFP cell mixing, the cells expressing the antibody-targeted membrane protein were pre-incubated for 45 min at 25°C with nivolumab (HY-P9903; CliniSciences), atezolizumab (HY-P9904; CliniSciences), or cetuximab (HY-P9905; CliniSciences) antibodies at concentrations ranging from 2 ng/ml to 0.1 mg/ml, in their culture medium. Binding between the two types of cells was estimated using flow cytometry (CytoFLEX; Beckman Coulter) as the percentage of double-positive events (GFP$^+$ RFP$^+$) among transfected cells, or among RFP$^+$ cells when transfection efficiency of RFP-positive cells was the limiting factor because of toxicity of over-expressed membrane proteins. Cytometry data analysis was performed on CytExpert (Beckman Coulter) or FlowJo (BD Biosciences) software.

#### Cell assay between Raji and 293-F cells

293-F cells stably expressing anti-CD19 or anti-EGFR CAR, and GFP were seeded as previously described in 12-well low-adhesion plates. Raji cells were cultured in six-well plates. 24 h post–293-F seeding, Raji and 293-F cells were counted. Raji cells were incubated in PE-conjugated anti-CD20 (302306; BioLegend) diluted at 1:20 in PBS/2% FBS, for 30 min at RT. After washing in FreeStyle 293 Expression Medium, cells were resuspended in the same medium and a co-culture at a ratio of 1:1 (Raji:293-F) was performed for 1–2 h at 25°C. Data were analyzed as mentioned above.

#### Cell assay between K562 and 293-F cells

293-F cells were seeded as previously described in 12-well low-adhesion plates and transfected to express HLA-G receptors and RFP. K562 cells stably expressing GFP and HLA-G isoforms were cultured in six-well plates. 24 h post–293-F transfection, K562 and 293-F cells were counted and a co-culture at a ratio of 1:2 (K562:293-F) was performed for 1–2 h at 25°C. Data were analyzed as mentioned above.

#### Cell assay between WOP and 293-F cells

293-F cells were seeded as previously described in 12-well low-adhesion plates and transfected to express HLA-G receptors and RFP. 6 × $10^5$ cells and WOP cells stably expressing HLA-G isoforms were seeded in 6-cm culture dishes the day before transfection. They were then transfected with a GFP expression vector using jetPRIME transfection reagent according to the manufacturer's protocol. 24 h post-transfection, WOP cells were detached with 10 mM EDTA in PBS, washed, and resuspended in PBS supplemented with 2% FBS. WOP and 293-F cells were counted, and a co-culture at a ratio of 1:2 (WOP:293-F) was performed for 1–2 h at 25°C. For the competition assay with anti-LILRB1 inhibitory antibody (20G10), WOP cells that were used stably expressed HLA-G1, $\beta$2-microglobulin, and GFP. Before GFP and RFP cell mixing, the cells expressing the antibody-targeted membrane protein were pre-incubated for 45 min at 25°C with anti-LILRB1 antibody at concentrations ranging from 2 ng/ml to 0.2 mg/ml. Data were analyzed as mentioned above.

### cDNA library screening and subcloning

The human ORFeome library v8.1 (12407 clones) (CCSB Human ORFeome Collection, Dana-Farber Cancer Institute, Boston, MA, USA) implemented with the ORFeome Collaboration Collection (OCC) (2188 clones) contains 14,595 human cDNAs representing about 12,000 human genes (Yang et al, 2011; ORFeome Collaboration et al, 2016). The cDNAs are grouped in 163 pools of 87 ORFs each on average (a gift from Y. Jacob). For mammalian expression, the library was subsequently transferred into a phCMV-DEST-A vector (Mousseau et al, 2024) (accepted manuscript). Briefly, the vector transfer was performed using the LR Clonase II enzyme mix (Thermo Fisher Scientific) following the manufacturer's protocol. The reaction product was then electroporated into electrocompetent DH5$\alpha$ Escherichia coli. Approximately 5,000 colonies resulting from the transformation of each pool were harvested, and plasmid DNA was extracted using the NucleoSpin kit (740588.250; Macherey-Nagel) to reconstitute the 163 pools with a concentration adjusted to 100 ng/$\mu$l.

The cell interaction assay was performed with each of these pools as described above and in Fig 3A. Positive pools were used to transform competent DH5$\alpha$ E. coli (Thermo Fisher Scientific). Colonies were grown separately and then pooled to generate 16 subpools of 12 ORFs each. These subgroups were analyzed by the cell interaction assay, and the 12 individual ORFs of positive subgroups were then tested. The plasmids encoding positive candidates were sequenced with phCMV forward (fwd) and reverse (rev) primers (see Table 3), and sequences were aligned on the human genome using Blastn (NCBI, Homo sapiens).

### Production of soluble proteins

293-F cells were seeded at $10^6$ cells/ml in six-well low-adhesion plates. The day after, they were transfected with plasmids encoding soluble Fc- or His-tagged receptors using FectoPro transfection reagent and following the manufacturer's protocol. Protein production was allowed for 4 d, then cells were collected and centrifuged for 4 min at 400$g$, and the supernatant was filtered through a 0.22-$\mu$m filter. Supernatants were stored for several weeks at 4°C, and protein production was verified by Western blot.

### Binding assay

For binding assays, WOP cells were detached using 10 mM EDTA in PBS and washed once in PBS/2% FBS, and $10^5$ to 2 × $10^5$ cells per well

were transferred in 96-well plates. For non-adherent cells, 293-F or K562 cells were counted and $10^5$ to $2 \times 10^5$ cells per well were transferred in 96-well plates and washed once. All washing steps and dilution are performed in PBS/2% FBS. Cells were incubated in 50 $\mu$l soluble receptors containing the supernatant for 1 h on ice. Samples were washed twice and then incubated for 45 min, on ice, in 50 $\mu$l fluorophore-conjugated antibodies (Goat anti-Human IgG H+L Alexa Fluor 488 Conjugate, A-11013; Invitrogen; or Mouse Penta-His Alexa Fluor 647 Conjugate, 1019252; QIAGEN) diluted at 1:500 in PBS/2% FBS. Cells were then washed twice, and data were acquired with CytoFLEX instrument and analyzed with either CytExpert or FlowJo software.

### Statistical analysis

Data visualization was performed in Excel for bar graphs or in GraphPad Prism 10 for dose–response curves. In all bar graphs and dose–response curves, data are means ± SEM from at least three independent experiments. Statistics were performed using a paired $t$ test with GraphPad Prism. The statistical details for each graph, including the number of independent experiments and statistical significance, are indicated in the figure and/or figure legends. For dose–response curves, $IC_{50}$ values were assessed using GraphPad Prism.

## Data Availability

All data generated or analyzed during this study are included in this article and its supplementary information files, or can be available from the corresponding author upon request.

## Supplementary Information

## Acknowledgements

We thank the Imaging and Cytometry Core Facility (PFIC) (Unit AMMICa, Gustave Roussy) for the expertise and advice in using cytometry instruments and for cell sorting, Maude Marchais and Marianne Mangeney for CAR lentiviral vectors, and all the members of UMR9196 laboratory for helpful discussions, for plasmids, and for cells. T Aymoz-Bressot has been awarded a PhD grant from Paris-Saclay University and the French Ministry of Higher Education and Research, and received a fourth-year thesis grant from La Ligue Contre le Cancer.

### Author Contributions

T Aymoz-Bressot: conceptualization, formal analysis, validation, investigation, visualization, methodology, and writing—original draft, review, and editing.
M Canis: formal analysis, validation, and investigation.
F Meurisse: resources and writing—review and editing.
A Wijkhuisen: resources and writing—review and editing.
B Favier: resources and writing—review and editing.
G Mousseau: resources and writing—review and editing.
A Dupressoir: resources, supervision, and writing—review and editing.
T Heidmann: resources, supervision, and writing—review and editing.
A Bacquin: conceptualization, formal analysis, supervision, validation, investigation, visualization, methodology, and writing—original draft, review, and editing.

### Conflict of Interest Statement

A Bacquin and G Mousseau are employees of VIROXIS. T Heidmann is President of VIROXIS. B Favier and A Wijkhuisen are co-inventors on a patent related to the anti-LILRB1 antibody. The authors have no other competing financial interests.

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
