## [Reviewer comments · Life Science Alliance]

Cell-Int: A cell-cell interaction assay to identify native membrane protein interactions

Thibaud Aymoz-Bressot, Marie Canis, Florian Meurisse, Anne Wijkhuisen, Benoit Favier, Guillaume Mousseau, Anne Dupressoir, Thierry Heidmann, and Agathe Bacquin

DOI: <https://doi.org/10.26508/lsa.202402844>

Corresponding author(s): *Agathe Bacquin, Viroxis*

Review Timeline:	Submission Date:	2024-05-27
	Editorial Decision:	2024-05-28
	Revision Received:	2024-08-21
	Editorial Decision:	2024-08-23
	Revision Received:	2024-08-29
	Accepted:	2024-08-29

Scientific Editor: *Eric Sawey, PhD*

Transaction Report:

Please note that the manuscript was previously reviewed at another journal and the reports were taken into account in the decision-making process at Life Science Alliance.

May 28, 2024

Re: Life Science Alliance manuscript #LSA-2024-02844-T

Dr. Agathe Bacquin
Viroxis
Gustave Roussy
39 rue Camille Desmoulins
Villejuif 94805
France

Dear Dr. Bacquin,

Thank you for submitting your manuscript entitled "A cell-cell binding assay to identify native membrane protein interactions" to Life Science Alliance. We invite you to submit a revised manuscript addressing the following Reviewer comments:

- Address Reviewer 2's comments.
- Address Reviewer 3's points # 1, 2, 3, 5, 6, 7, 9, 10, 11, 14, 15, 16, 19-28.

Thank you for this interesting contribution to Life Science Alliance. We are looking forward to receiving your revised manuscript.

Sincerely,

- A letter addressing the reviewers' comments point by point.
- An editable version of the final text (.DOC or .DOCX) is needed for copyediting (no PDFs).
- High-resolution figure, supplementary figure and video files uploaded as individual files: See our detailed guidelines for preparing your production-ready images, <https://www.life-science-alliance.org/authors>
- Summary blurb (enter in submission system): A short text summarizing in a single sentence the study (max. 200 characters including spaces). This text is used in conjunction with the titles of papers, hence should be informative and complementary to the title and running title. It should describe the context and significance of the findings for a general readership; it should be written in the present tense and refer to the work in the third person. Author names should not be mentioned.
- By submitting a revision, you attest that you are aware of our payment policies found here: <https://www.life-science-alliance.org/copyright-license-fee>

B. MANUSCRIPT ORGANIZATION AND FORMATTING:

Re : Life Science Alliance manuscript #LSA-2024-02844-T

Thank you for processing our manuscript #LSA-2024-02844-T by Aymoz-Bressot et al. As requested, you will find below our point-by-point response (in black) to the following reviewer comments (in blue):

- Reviewer 2's comments.
- Reviewer 3's points # 1, 2, 3, 5, 6, 7, 9, 10, 11, 14, 15, 16, 19-28.

We would like to thank the reviewers for reviewing our manuscript, and providing valuable feedback on how we can improve this work.

A revised version has now been uploaded to the LSA Submission site, with appropriate changes that address the reviewers' concerns. The line numbers of the revised manuscript, where the revisions were made, are specified in the corresponding responses below.

Reviewer #2 (Comments to the Authors (Required)):

This manuscript by Aymoz-Bressot et al describes a method to track and screen interactions between membrane proteins expressed in HEK-293 cells. Read-out is by flow cytometry through mixing of fluorescent reporters expressed in the two interacting cells. The authors provide convincing evidence that the approach is functional for a variety of applications including different types of transmembrane proteins, screening approaches to identify binding partners as well as inhibitor studies. While there is no fundamental criticism on the data, it is clear that this approach of overexpressing binding partners in HEK cells is far from being physiologically relevant. This refers to the chosen cell type as well as the high protein density on the cell surface. My main criticism refers to this not being clearly stated and discussed. The limitations of this method should be listed in a separate paragraph of the Discussion without extending its overall length. This discussion should also include clear statements about the sensitivity of the approach.

We are grateful to the reviewer for his careful reading of our manuscript. We concur with the restriction regarding the overexpression of binding partners, which is not a physiological state. Indeed, supplementary experiments were conducted to compare overexpression in transfected or transduced cell lines with physiological protein expression. The number of protein molecules expressed on the cell surface was determined using the Quantibrite quantification kit, which allows for the quantification of membrane proteins with PE-coupled antibodies (Serke et al., 1998). Specifically, we conducted a comparative analysis of HLA-G expression levels between cells that had been transfected or transduced with an HLA-G-expressing vector and JEG-3, a choriocarcinoma cell line derived from placental tissue, which is known to express HLA-G endogenously. The results demonstrated that the transfected 293-F and wild-type JEG-3 cells, respectively, expressed 33,000 and 24,000 copies of HLA-G at their surface, indicating that they were in the same order of magnitude. However, WOP cells stably expressing HLA-G were found to have approximately 297,000 membrane copies of HLA-G1, which is twelve times more than that observed in JEG-3 cells. Similarly, when comparing 293-F cells transfected with a PD-1 or PD-L1 expression vector to peripheral blood mononuclear cells (PBMCs), we also

measured approximately ten times more extracellular PD-1 and PD-L1 molecules in the transfected cells. These findings suggest that our approach assays membrane protein interaction with non-physiologically overexpressed proteins, which are generally well above the endogenous level.

To address the reviewer's comment, we have now incorporated a table (Table 2) into the discussion section of the manuscript. The table provides a comparative analysis of the current methods for screening new membrane protein interactors, including our own approach. The table describes the characteristics of each technique, including considerations regarding protein expression, conformation, microenvironment, and potential limitations. In particular, it indicates whether the techniques utilize the endogenous level of expression of membrane proteins or not. Furthermore, the table indicates that our technique may have sensitivity limitations. For instance, the HLA-G1/CD8 Kd value of 72 μ M represents a potential sensitivity threshold for the Cell-Int technique, as no interaction signal was observed for this pair. Table 2 and our technique's limitations are discussed on lines 436-456 and 388-389.

Minor issues

Typos: lines 506, 509, 594 and 595 check the numbers

The manuscript has been corrected accordingly at lines 584, 588, 679, and 680.

Fig. 1C and 2A: The bars cannot be distinguished with the chosen color code.

The color code has been updated to provide a clear distinction between each condition.

Fig. 2B: check color code seems to be wrongly labeled

The color code has now been corrected.

Fig. 4B: wrong color code for empty vector bar?

The color code has now been corrected.

Reviewer #3 (Comments to the Authors (Required)):

In this manuscript, the authors introduce a new technique to study intercellular protein-protein interactions. This technique presents a straightforward, experimentally simple way to assay binary protein-protein interactions at the cell surface, and the authors show this technique applied to several protein pairs including known interacting pairs to validate the technique, investigation of uncharacterized or debated protein pairs, as well as a screen to identify interactors in a more unbiased manner. The authors also demonstrate the utility of this technique to assay the efficacy of PPI modulators such as blocking antibodies.

As the authors discuss in their introduction, cell surface proteins comprise a large proportion of therapeutic targets and they will remain forever relevant for disease biology and drug discovery. Therefore, having robust techniques that allow facile study of receptor-ligand and other cell surface protein-protein interactions is essential for the field. While there are many existing techniques that address this important biology, the method presented by the authors in this manuscript allows investigation of these protein-protein interactions in arguably one of the closest to physiological states. Due to the binary nature of the assay and its reliance on a flow cytometry-based workflow, I am skeptical that it has the necessary throughput or sensitivity for truly rapid, unbiased novel receptor-ligand/PPI discovery or high-throughput screening of modulatory molecules. Rather, this technique appears to be more amenable for validation of receptor-ligand/PPI pairs or efficacy of modulatory molecules discovered elsewhere. While this limits the scope of the technique, it is attractively simple and would still be quite useful if it can be widely and easily applied to any protein pair of interest - however, this is still in question for me after reading the manuscript. There are additional experiments, controls, experimental details, and discussion points to be addressed by the authors that will help demonstrate the validity of the technique presented and importantly, its range of applicability. Therefore, I cannot recommend this manuscript for publication in its current state but will be happy to review again once the following points are addressed:

We are very grateful to the reviewer for his comprehensive and insightful analysis of our work and the valuable feedback he provided. Please find below our responses to the points raised in remarks #1, 2, 3, 5, 6, 7, 9, 10, 11, 14, 15, 16, 19-28, as requested by the editor.

1. In the introduction, the authors do a good job of discussing relevant biological systems that highlight the importance of intercellular cell surface protein-protein interactions, however I would also like to see more discussion of specific existing techniques that are currently used to study these interactions and how their pros/drawbacks compare to the technique presented by the authors. Relevant techniques that come to mind are proximity labeling (Loh et al. Cell 2016, Takano et al. Nature 2020, Geri et al. Science 2020), AVEXIS (Martinez-Martin et al. Cell 2018), RDIMIS (Cao et al. PNAS 2021), uLIPSTIC (Nakandakari-Higa et al. Nature 2024), and split complement systems (Martell et al. Nature Biotech 2016).

In response to the reviewer's comment, we have incorporated a table (Table 2) into the discussion section of the manuscript. This table compares current methods for screening new membrane protein interactors, including those mentioned by the reviewer and our own approach. It outlines the characteristics of each technique, such as considerations regarding protein expression, conformation, microenvironment, and potential limitations. Associated discussion on these approaches is also included (lines 436-456).

2. It is unclear what expression level of the proteins being assayed is required for interaction and aggregation in the cell interaction assay. Please approximate what number of protein molecules on the cell surface that is required to induce aggregation, and quantify the correlation between number of cell surface molecules (by titrating expression) and double-positive events. Ideally this should be done for several protein pairs that have a broad range of known binding affinities in order to understand the general limits and requirements for this assay. In addition, each protein assayed in this study should include cell surface expression data as in Fig. S1D when possible.

To address the reviewer's comment, we conducted additional experiments to further characterize the expression of binding proteins at the cell surface that are necessary for the interaction between two cell populations to be detected using Cell-Int (Figure S6). The number of protein molecules expressed on the cell surface was determined using the Quantibrite quantification kit, which allows for the quantification of membrane proteins with PE-coupled antibodies (Serke et al, 1998). We performed the cell interaction assay between 2 binding pairs, HLA-G1/LILRB1 and PD-1/PD-L1, with serial dilutions of LILRB1 and PD-L1, along with the quantification of surface proteins and transfection efficiency (Figure S6). The interaction signal between LILRB1 and HLA-G1/ β 2m was increased in comparison to the e.v. control condition at dilutions of 1 to 1/300th of the LILRB1 expression vector, although the differences were only statistically significant at dilutions of 1 and 1/10th. With regard to the PD-1/PD-L1 pair, the interaction signal was found to be significantly elevated at dilutions ranging from 1 to 1/300th (Figure S6B, left panel), as with the Juno/Izumo pair (Figure 1E). When the expression of the proteins was studied in parallel, the interaction signals of both the HLA-G1/LILRB1 and PD-1/PD-L1 pairs exhibited a stronger correlation with transfection efficiency than with the number of proteins expressed at the cell surface (Figure S6, right panels). In particular, our results indicate that below a transfection efficiency of 20%, it may be challenging to detect cell-cell interactions using Cell-Int. It suggests that transfection efficiency is the primary factor influencing the sensitivity of the technique, while protein copy number at the cell surface may act as a secondary limiting factor.

This analysis has now been incorporated into the result section (lines 326-341) with the associated Figure S6. Furthermore, in accordance with the reviewer's request, we have included, as much as possible, cell surface expression data for the proteins assayed in this study (Figs S2, S3 and S5).

3. It is unclear how many cells are involved in the PP-induced cell aggregates - are all cells in the double-positive aggregates bearing interacting protein pairs, or do the aggregates start attracting cells non-specifically once they begin to form? Please provide additional FSC and SSC flow plots, e.g. FSC-A x FSC-

h, to have a better understanding of the size of the aggregates formed and please distinguish what size of cell aggregates you are gating in your assay (doublets? Larger aggregates?). If you collect and dissociate these double-positive aggregates, how many cells are not expressing either overexpressed protein and are the number of cells expressing either protein being assayed equal?

In response to the reviewer's comment, we present in the figure below the SSC-A x FSC-A and FSC-A x FSC-H plots obtained for the Juno/Izumo1 pair. As can be observed in the FSC-A x FSC-H plots, the percentage of single cells is lower for the Juno x Izumo1 condition in comparison to the background conditions, which is consistent with the increase in aggregates. As we could not easily distinguish doublets from larger aggregates in FSC-A x FSC-H plots, we conducted additional experiments to further characterize the aggregates. In particular, we employed two complementary approaches and applied them to the Juno x Izumo1 aggregates. The first approach involved flow cytometry analysis of double positive events following Hoechst staining. Multiple peaks were observed. For each peak, the mean fluorescence intensity was divided by that of the singlets, thus enabling the identification and approximation of the frequency of doublets, triplets, quadruplets, and higher aggregates. The results have been included in the manuscript in Fig S1D and E, with an associated description in the results section, lines 156-161.

In the second approach, double positive events were sorted using a cell sorter into wells of 96-well plates. The sorted aggregates were then analyzed by fluorescence microscopy. Image analysis of over 100 aggregates allowed us to determine the frequency of each aggregate type, with the results presented in Fig S1E and F, and in the associated text, lines 161-164. We also examined whether all cells included in the aggregates were either GFP- or RFP-positive, noting that one-third of the aggregates contained one or more non-fluorescent cells. This observation has been included in the manuscript, lines 164-166.

Figure related to remark #3: FSC-H x FSC-A plots show the presence of large aggregates

Top: FSC-A x SSC-A plots showing the gating strategy to include large cell aggregates, for the 4 coculture conditions tested: e.v x e.v.; e.v x Izumo; Juno x e.v.; and Juno x Izumo. Bottom: FSC-A x FSC-H plots showing the presence of doublets and larger aggregates. Single cells are gated and the percentage in the gate is indicated.

5. Please provide an extended discussion of the relationship between protein-protein interaction affinity and the rate of double-positives in your cell interaction assay, including why there is not a correlation between affinity and cell aggregate formation, and what you believe the affinity range of your assay to be and why. Please also include a complete reporting of affinity constants in Table 1 for each protein pair assayed, and if not previously known please indicate that in the table.

As illustrated in the graph below, which depicts interaction signals measured by Cell-Int technique and corresponding Kd values (when available in the literature), there is indeed no correlation between affinities and cell aggregate formation.

In accordance with the aforementioned discussion regarding remark 2, it is likely that this absence of correlation is attributable, at least in part, to the dependence of the interaction signal on the level of transfection and the quantity of proteins expressed at the cell surface. Furthermore, conformational differences between soluble ectodomains (which are utilized in most techniques for assessing

affinities) and full membrane-embedded proteins (which are studied using Cell-Int) are likely to result in variations in affinity. Concerning the affinity range of our assay, we are capable of measuring interactions between binding pairs exhibiting affinities spanning from the nanomolar to the micromolar range. The HLA-G1/CD8 Kd value of 72 μ M represents a potential sensitivity threshold for the Cell-Int technique, as no interaction signal was observed for this pair.

In order to address the reviewer's concern, we have now proposed this potential sensitivity threshold in the manuscript and indicated that the interaction signal measured by Cell-Int does not correlate with affinities (lines 386-396). Furthermore, we have completed Table 1, which depicts the affinities of all interacting pairs that were tested in this study.

Figure related to remark #5: % double positive x Kd plot

The graph depicts the interaction signals as a function of the corresponding Kd values (when available) of the studied interacting pairs.

6. Please discuss if the kinetics and temperature for the assay need to be optimized for each protein pair assayed, and if not, why these parameters would not change depending on the affinity and on/off rates for each interaction.

In order to address the reviewer's concern, we checked whether the selected parameters were optimal for additional protein partners. To this end, we examined the kinetics of the cell interaction signal and the impact of temperature on the following interactions: PD-1/PD-L1, GLN Env/Slc19A1, HLA-G1/LILRB1 and HLA-G1/LILRB2 (see Figure below). The conditions that were optimal for the Juno/Izumo1 interaction analysis (i.e. 1 to 2 hours at 25°C) were also found to be optimal for all the other pairs tested. It is noteworthy that at either 25°C or 37°C, the percentage of double positives reaches its maximum, with the exception of the GLN Env/Slc19A1 pair, where the interaction signal appears to exhibit a slight decrease at 37°C in comparison to 25°C. However, at 37°C, a higher background interaction is observed for all tested pairs, indicating that non-specific interactions are more likely to occur at this temperature than at 25°C. At 4°C, the results are more variable between pairs, with PD-1/PD-L1 showing a high and significant interaction signal, GLN Env/Slc19A1 and HLA-G1/LILRB1 showing a weak to moderate but significant interaction, and HLA-G1/LILRB2 showing only a weak and partially significant interaction. Therefore, 4°C does not appear to be an optimal temperature for the assay. With regard to the kinetics of the cell interaction, as observed for Juno/Izumo, the cell interaction signal increases over time and reaches a plateau at 30 minutes to 2 hours of incubation.

In conclusion, these results indicate that the temperature and kinetics for conducting the cell interaction assay do not require optimization for each interacting pair under study. Furthermore, the conditions selected based on the Juno/Izumo interaction analysis appear to be suitable. This observation has now been incorporated into the first result section (lines 144-145 and 153-154).

Figure related to remark #6: Impact of kinetics and temperature on the cell interaction signal of PD-1/PD-L1, GLN Env/Slc19A1, HLA-G1/LILRB1 and HLA-G1/LILRB2

Quantification of the cell-cell interaction using Cell-Int is represented on bar graphs displaying the percentage of double-positive events among RFP positive cells (means \pm SEM). Statistics were performed using paired t-test (three to four independent experiments; ns, $P > 0.05$; *, $P \leq 0.05$; **, $P \leq 0.01$; ***, $P \leq 0.001$; ****, $P \leq 0.0001$). Three different temperatures (left panel) were tested for the co-incubation of the two cell batches for one hour: 4°C, 25°C, and 37°C. Cell interaction signal kinetics were followed (right panel) by co-incubation of the two cell batches for up to three hours at 25°C, with incubation times represented as shades of gray, from 10 min (white) to 3 hours (black).

7. Why are the fluorescent protein markers and proteins of interest not on the same plasmid, e.g. co-expressed on different plasmids and not one on bi-cistronic plasmid, or better yet under the same promoter with an IRES or 2A? I understand this would reduce the modularity, but having completely separate plasmids co-transfected leads to more indirect data that is more difficult to interpret.

The use of a single expression vector to express both the protein of interest and the fluorescent protein marker has indeed the potential to enhance the correlation between the percentage of fluorescent cells and the percentage of cells expressing the protein of interest, thereby improving the sensitivity of the technique. To investigate this proposition, we conducted a comparative analysis of the interaction signal between Juno and Izumo1 or Izumo1-IRES-RFP (see Figure below). The interaction signals were very similar, indicating that the expression of both the fluorescent protein and the protein of interest in a single vector does not necessarily yield enhanced outcomes. However, it is possible that this observation does not extend to all proteins of interest.

Figure related to remark #7: Interaction signals between Juno and Izumo or Izumo-IRES-RFP detected by Cell-Int.

Juno/Izumo1 and Juno/Izumo1-IRES-RFP interactions were assessed using Cell-Int by transfecting one batch of 293-F cells (in green) with GFP and Juno expression vectors or an empty vector control (e.v.), and a second batch (in red) with RFP and Izumo1 expression vectors or an e.v. control, or Izumo1-IRES-RFP only. Quantification of the cell-cell interaction is represented on bar graphs displaying the percentage of double-positive events among transfected cells (means \pm SEM). Statistics were performed using paired t-test (three independent experiments; *, $P \leq 0.05$).

Besides, as mentioned by the reviewer, the use of an IRES-RFP construct would also reduce modularity as it would require additional molecular cloning into appropriate vectors.

In response to the reviewer's comment, we have amended the text to include this information in the first result section (lines 166-170).

9. It is unclear what is actually being assayed in Fig. 2c - are these MeVH-hEGF proteins? A figure showing the domain map for the fusion constructs would be useful. If so, a control with MeVH only should be included. Also, again, seeing data in EGFR KO or KD cells.

To clarify notations, we have now abbreviated the Measles virus hemagglutinin as H and the fusion protein combining H with EGF as H-EGF. A schematic representation of both forms was also included in the supplementary figures (Fig S3K). As requested by the reviewer, a control condition examining the interaction between H only and EGFR (Fig S3O) was added. This shows, as expected, that the interaction between H-EGF and EGFR is due to EGF and not H. As suggested by the reviewer, a further experiment was conducted to ascertain the source of the background interaction signal observed in the control condition with H-EGF alone. This background signal was effectively eliminated when interaction with endogenous EGFR was inhibited by the use of the anti-EGFR inhibitory antibody Cetuximab (Fig S3P).

Similarly, we conducted an additional experiment to analyze the interaction signal between 293F cells expressing Syncytin-A and 293T cells that were knocked out for Ly-6E endogenous expression, compared with wt 293T cells (Fig S3J). As anticipated, our findings confirmed that Syncytin-A-induced background interaction is effectively eliminated when depleting endogenous Ly-6E expression.

The second result section has been updated to include the aforementioned observations (lines 209-212 and 222-225).

10. For the ORFeome screen in Fig. 3, please discuss why iterative approach was used and not cell sorting and sequencing and how this relates to the sensitivity of your assay.

This could indeed be an effective alternative method for identifying the candidate responsible for the formation of the aggregates. However, due to background interactions, numerous false-positive proteins would be sequenced, making it challenging to identify the specific positive candidate among the false-positives. Additionally, to obtain sufficient material for sequencing, a substantial number of aggregates would need to be collected. The iterative approach allows for the purification of the candidate, making it more efficient to sequence, as only the single plasmid used for transfection is sequenced in the final round. Furthermore, this approach was selected due to its time and cost effectiveness. We have now included a remark to this effect in the manuscript (lines 241-242).

11. Was LILRB2 or other positive control interactors present in one of the pools for the ORFeome screen in Fig. 3? If so, please discuss why only LILRB1 was found through screening the pools and how this relates to the sensitivity of your assay.

LILRB2 was used as a positive control for the screen, given its absence from the Orfeome library. This information has now been incorporated into the manuscript (lines 248-249). KIR2DL4, CD160, and CD8a are present in the library, in pools 81, 27, and 143 respectively. However, these pools did not result in an interaction with HLA-G during the screening process. This finding is consistent with the results observed in Fig 4D and Fig 5, which demonstrated that even purified receptors KIR2DL4, CD160, and CD8a did not interact with HLA-G.

14. Please comment on why you think the other receptors (KIR2DL4, CD160, CD8) did not lead to an interaction. E.g. do you predict that the interactions are too weak, or too transient, or that they require additional cofactors?

As discussed in a paragraph of the discussion (lines 474-491), the remaining three receptors described for HLA-G are significantly less characterized. We only know a K_d value for HLA-G:CD8 which is $72\mu\text{M}$. This markedly reduced affinity in comparison to LILRB1 and LILRB2 may account for the absence of a signal observed through our technique, suggesting a potential sensitivity threshold for the technique itself. However, it is also possible that the lack of signal is due to the requirement for a cofactor or even insufficient CD8 membrane expression. Indeed, when Fig S5 is considered, it can be seen that CD8 is the least expressed receptor among all HLA-G receptors, with lower fluorescence intensity and percentage of transfected cells than the other receptors. Regarding KIR2DL4, only one research group has described its interaction with soluble HLA-G which was found to occur in endosomes. Here, despite KIR2DL4 being well expressed on the plasma membrane, we cannot exclude that a cofactor involved in the endocytic pathway and required for the interaction to occur is absent in the 293-F cells. Additionally, it is possible that soluble HLA-G would contact residues of KIR2DL4 that are not accessible to membrane HLA-G. Finally, the specific endocytic environment could be required for the interaction to occur. As for CD160, little evidence is found in the literature, and the interaction is presumed to have a weak affinity, given that MHC crosslinking markedly enhanced the interaction.

15. Please discuss why in Fig. 4b, LILRB1 induced double-positive events without HLA-G1/B2m expression (empty vector), but in Fig. 4d there was minimal binding of the ectodomain-Fc, if presumably the background binding before was due to binding of endogenous substrates. Also, why do you think the ectodomain only gave such lower binding (%positive vs % double positive) than the cell-based assay?

Our primary hypothesis is that soluble tagged proteins do not necessarily behave like intact membrane proteins in their plasma membrane environment. In the case of Fc-tagged soluble LILRB1, it is conceivable that a distinct conformation could result in a reduction in affinity to CMH-I proteins, outside HLA-G. This hypothesis was added in the manuscript (lines 289-291). It is worth noting that this reinforces the advantage of Cell-Int being more physiologically relevant, as it analyzes the protein in its complete, intact form. In addition, while non-transfected cells are excluded from the analysis in the cell interaction assay, this is not the case in the binding assay. Therefore, the maximum binding is

limited by the transfection efficiency. This may contribute to both the lower background of LILRB1 and its lower binding to HLA-G1 in the binding assay relative to the cell-based assay.

16. Regarding the comment, "We noted a high background interaction of this soluble receptor with 293-F cells, supposedly due to binding to other endogenously expressed HLA class I molecules. This might explain the absence of differential binding of LILRB2-Fc between e.v and HLA-G1/ β 2m transfected 293-F cells." Why would this give different results than the cell-based assay in Fig. 4b where these endogenously expressed HLA class I molecules would still be present?

In line with the previous point, a slight alteration in the conformation of the LILRB2 ectodomain may also occur when produced as a soluble Fc-tagged form. This could potentially lead to differential binding to CMH-I proteins in comparison to the full intact membrane protein within its plasma membrane environment. This hypothesis has been added to the manuscript (lines 286-289).

19. In Fig. 5c-d, why did you not include a WOP empty vector control? Also, please comment on why it appears that you have some binding with LILRB2 in the cell-based assay in Fig. 5c when HLA-G1 or β 2m are expressed alone, and why this is not recapitulated in the ectodomain experiment in Fig. 5d.

We reasoned that in the case of murine WOP cells, which do not express human β 2m (in contrast to human cell lines 293-F and K562), β 2m stably transduced WOP cells were a superior control. β 2m is cloned into the same vector as the empty vector control and allows to exclude any potential interaction mediated by β 2m only. This ensures that the signal we measure is driven by HLA-G interaction with its receptors. This choice has now been justified in the manuscript (lines 304-306)

In Fig 5C, the increase in signal is not statistically significant. The variability observed in this condition is likely due to LILRB2 toxicity when overexpressed, a phenomenon that we often encountered. Since the binding assay does not rely on transfection of membrane LILRB2, this issue is not present in this test.

20. In Fig. 6, why did you not include negative controls for cetuximab and anti-LILRB1 20G10, e.g. PD-1/PD-L1/2? Or generally for all panels, other pairs including the target but not the matched ligand e.g. EGFR x PD-1 for cetuximab?

We agree that the above-mentioned controls are lacking. Consequently, we analyzed the impact of anti-LILRB1 20G10 on the HLA-G1/LILRB2 interaction, as well as the effect of Cetuximab on the PD-1/PD-L1 interaction. As anticipated, no inhibitory activity was observed in either condition. Fig. 6C and D have been updated to include the aforementioned control conditions, and the associated text in the result section has been completed (lines 355-357 and 362-364).

21. Please comment on how the calculated IC50s from Fig. 6 compare to previously determined IC50 values.

We agree that this is an important point to consider. We have now included a new table in the discussion section (Table S1 and lines 416-419), which lists the IC50 values determined by Cell-Int, in comparison to previously reported values. For Nivolumab and Atezolizumab, the IC50 values determined in this study are somewhat higher than those reported in previous studies. However, they remain within the same order of magnitude, ranging from 2.7 to 5.6-fold higher. The discrepancies are likely attributable to differences in the techniques employed and, in particular, the form of the ligands (soluble versus membrane-bound). For Cetuximab, the authors of the mentioned study employed an in vitro cell toxicity assay and reported a high variation of IC50 values depending on the sensitivity of the cell line used. Furthermore, the authors investigated the interaction between EGF and EGFR, whereas we focused on H-EGF/EGFR, which is much less characterized and for which, to our knowledge, no IC50 value is reported. Regarding the anti-LILRB1 20G10, our study is the first to analyze this antibody.

22. Regarding the lack of binding for high sequence similarity proteins, e.g. PILRa vs PILRb binding to NPDC-1, are these patterns of binding known or previously reported? The discussion makes this sound

like a new discovery. Are there previously reported or predicted structures of the binding interface or conformation that provide some rationale for the differences in binding?

A review of the literature on paired receptors involved in the immune response indicates that there is no systematic sharing of ligands between activating and inhibitory receptors (Kuroki et al., 2012). In fact, inhibitory receptors have been found to bind to a greater number of ligands.

To the best of our knowledge, there is no existing literature demonstrating the absence of interaction between PILR β and NPDC-1. The study by Lu et al., published in PNAS in 2014, compared the structures of PILR α and PILR β . Their findings indicate that both proteins have a siglec-like fold and are capable of interacting with sialic acid. However, they have identified differences in sialic acid affinity involving specific residues. Using the example of HSV-1 glycoprotein B (gB)/PILR α interaction, they demonstrate that mutation of a single PILR β residue restores interaction with gB. It would be interesting to conduct similar structural studies on the PILR α /NPDC-1 interaction to determine whether it is also possible to identify the PILR α residues involved in the interaction and why this interaction is not detected with PILR β .

In order to address the reviewer's concern, we have expanded the discussion section on this matter (lines 397-410).

23. Please add extended discussion on the generalizability of is this technique really. Can it be expanded to more relevant cell models where overexpression may be difficult? What are the drawbacks of using a less physiologically/disease-relevant system like 293s, e.g if co-factors required for binding are not expressed or present in those cells? How much fine tuning is required for each pair? Can this really be used in a discovery/screening-manner or would factors like expression level, temperature, incubation time, cell density need to be fined tuned every time?

As indicated in our response to remark 1, a table has been added to the manuscript which lists a number of techniques which can be used to assess membrane protein interactions (Table 2). The table presents the characteristics of each technique, including our own, and also states that in our system, the protein levels are indeed not physiological.

Regarding the potential for expanding our technique to additional relevant cell models, our study demonstrated the ability to stably overexpress proteins through lentiviral transduction, followed by efficient cell sorting to select transduced cells when necessary. This strategy allows for the convenient overexpression of a protein of interest in a variety of cell lines, including suspension-growing cells like K562 and adherent cells like WOP. If an interaction is anticipated to occur in a specific cell subtype, potentially due to the expression of a co-factor, Cell-Int can be applied to a cell line that mimics this cell type. It is worth noting that in some instances, we observed that endogenous expression of binding partners was sufficient to induce cell-to-cell interaction (see Fig S3J and S3P, and response to remark 9 for Ly6e/Syncytin-A and EGF/EGFR interactions). This indicates that in certain contexts, endogenous expression of one binding partner is sufficient to detect interaction with its overexpressed partner using Cell-Int.

Regarding the degree of fine-tuning necessary for each pair, we conducted kinetics and temperature experiments on four binding pairs (see response to remark 6 and associated figure) in addition to Juno/Izumo in Figure 1. Some differences can be observed between the results of the different binding pairs. However, the conditions that we selected based on Juno/Izumo (1 to 2 hours of incubation time at 25°C) yielded significant interaction for each pair with at least equivalent or sometimes better signals than the other conditions. These five examples suggest that no specific fine-tuning is required when investigating novel interactions.

24. Please provide citations for the techniques described in "...mass spectrometry-based techniques and loss-of-function screens using CRISPR/Cas9 technology. However, both types of approach are dependent on the level of endogenous expression and may miss weakly expressed receptors. Elisa-like in vitro screens using protein ectodomains are also used but they rely on the ability of a protein ectodomain to interact similarly to the original membrane protein and do not test interactions under physiological conditions."

The references have been incorporated into Table 2, which describes the commonly used techniques for assessing the interaction or proximity between membrane proteins. Additionally, they have been included in the associated paragraph (lines 436-456).

25. In the discussion, please expand on why you hypothesize that a more stabilized plasma membrane-expressed HLA-G1 would require $\beta 2m$ to bind LILRB2 but not the soluble version? Do you think that the soluble version is binding LILRB2 in a different manner?

As with the differential binding of LILRB1 and LILRB2 to CMH-I molecules when expressed as soluble ectodomains compared to full membrane-embedded proteins, it is indeed possible that HLA-G soluble version interacts in a different manner. In line with this, the study by Arns et al. (Arns et al., *Frontiers in Immunology*, 2020) using structural modeling suggests that the residues involved in the interaction of HLA-G with LILRB2 (and LILRB1) are less accessible in the membrane-bound HLA-G1 than in the soluble form. A functional study by Zhang et al. (Zhang et al., *Human Immunology*, 2014) demonstrated that the inhibitory effect of HLA-G5 (i.e. the soluble form equivalent to HLA-G1) on NK cell activation was more pronounced than that of HLA-G1. Our analysis of HLA-G1 plasma membrane expression in the presence or absence of $\beta 2m$ (see Fig S5B) indicates that HLA-G1 levels at the plasma membrane are increased when $\beta 2m$ is concomitantly overexpressed. This suggests that $\beta 2m$ plays a stabilizing role on plasma membrane HLA-G1. We have extended the discussion on this matter in the article (lines 462-473).

26. This claim needs a citation: "Indeed, soluble HLA-G was found to be internalized in endosomes in a process involving KIR2DL4."

The appropriate reference has been added (line 478).

27. Given this statement, "The weak binding measured in vitro might explain the absence of interaction detected by the cell-cell interaction assay in a physiological context," can you please comment in your discussion what is the affinity range that can be assayed with your technique?

This potential sensitivity limitation of our technique has now been discussed at lines 388-389, and indicated in the Table 2, which lists a number of techniques that can be used to assess membrane protein interactions, with an analysis of their respective advantages and disadvantages.

28. In the methods, please describe for the cell interaction assay what transfection efficiency is required, what expression level is required, how is expression level measured, what cell density required, what buffer are cells mixed together in, and what buffer and density are the cells analyzed in.

In accordance with the reviewer's request, we have expanded the technical details in the method section, lines 611-619.

Thank you for your work in developing and presenting this technique, I am looking forward to seeing your revised manuscript. Also, I suggest that you come up with a name for this new technique ;) best of luck to you.

In line with the reviewer's suggestion, we have named the cell-cell interaction assay developed here "Cell-Int." We would like to thank the reviewer once more for his comprehensive review of our study and all of his feedback, which we hope has enabled us to make significant improvements to the quality of the article.

August 23, 2024

RE: Life Science Alliance Manuscript #LSA-2024-02844-TR

Dr. Agathe Bacquin
Viroxis
Gustave Roussy
39 rue Camille Desmoulins
Villejuif 94805
France

Dear Dr. Bacquin,

Thank you for submitting your revised manuscript entitled "Cell-Int: A cell-cell interaction assay to identify native membrane protein interactions". We would be happy to publish your paper in Life Science Alliance pending final revisions necessary to meet our formatting guidelines.

- please be sure that the authorship listing and order is correct
- please upload table files as editable doc files

LSA now encourages authors to provide a 30-60 second video where the study is briefly explained. We will use these videos on social media to promote the published paper and the presenting author (for examples, see <https://docs.google.com/document/d/1-UWCfbE4pGcDdcgzcmiuJI2XMBJnxKYeqRvLLrLS08s/edit?usp=sharing>). Corresponding or first-authors are welcome to submit the video. Please submit only one video per manuscript. The video can be emailed to contact@life-science-alliance.org

A. FINAL FILES:

B. MANUSCRIPT ORGANIZATION AND FORMATTING:

Sincerely,

August 29, 2024

RE: Life Science Alliance Manuscript #LSA-2024-02844-TRR

Dr. Agathe Bacquin
Viroxis
Gustave Roussy
39 rue Camille Desmoulins
Villejuif 94805
France

Dear Dr. Bacquin,

Thank you for submitting your Methods entitled "Cell-Int: A cell-cell interaction assay to identify native membrane protein interactions". It is a pleasure to let you know that your manuscript is now accepted for publication in Life Science Alliance. Congratulations on this interesting work.

DISTRIBUTION OF MATERIALS:

Again, congratulations on a very nice paper. I hope you found the review process to be constructive and are pleased with how the manuscript was handled editorially. We look forward to future exciting submissions from your lab.

Sincerely,
